# Evolution shapes interaction patterns for epistasis and specific protein binding in a two-component signaling system

Zhiqiang Yan [1] & Jin Wang [2] ✉

The elegant design of protein sequence/structure/function relationships arises from the interaction patterns between amino acid positions. A central question is how evolutionary forces shape the interaction patterns that encode long-range epistasis and binding specificity. Here, we combined family-wide evolutionary analysis of natural homologous sequences and structure-oriented evolution simulation for two-component signaling (TCS) system. The magnitude-frequency relationship of coupling conservation between positions manifests a power-law-like distribution and the positions with highly coupling conservation are sparse but distributed intensely on the binding surfaces and hydrophobic core. The structure-specific interaction pattern involves further optimization of local frustrations at or near the binding surface to adapt the binding partner. The construction of family-wide conserved interaction patterns and structure-specific ones demonstrates that binding specificity is modulated by both direct intermolecular interactions and long-range epistasis across the binding complex. Evolution sculpts the interaction patterns via sequence variations at both family-wide and structure-specific levels for TCS system.

[1] Center for Theoretical Interdisciplinary Sciences, Wenzhou Institute, University of Chinese Academy of Sciences, Wenzhou, Zhejiang 325001, PR China.
[2] Department of Chemistry and Physics, State University of New York at Stony Brook, Stony Brook, NY 11790, USA. ✉email: jin.wang.1@stonybrook.edu

Proteins often perform functions through binding with their specific partners in the crowded cellular environment. Binding specificities between proteins are essential in precise recognition and avoiding crosstalks to highly similar competitors[1–4]. Protein binding, a more complex issue than protein folding, is dependent on the highly complicated nature of protein sequence/structure/function relationships[5–8]. The complexity of these relationships arise from the interaction patterns formed by the amino acid residues[9–13]. Anfinsen's thermodynamic hypothesis[14] suggested that structural prediction of proteins or protein complexes merely from their amino acid sequences is possible in theory. Recently, rapid advance of a wide range of methods from the interplay of physics, evolution and artificial intelligence has led to remarkable breakthrough in predicting protein structure[15–17]. However, the rule of interaction patterns that modulate specific binding remains elusive.

With the explosion of available homologous protein sequences, statistical analysis of multiple sequence alignment (MSA) has accelerated successful predictions of protein complex structures[15–21]. These methods exploited coevolution information of natural homologous sequences to extract direct contacts, as well as coupling dependencies which determine the long-range intramolecular or intermolecular communications between residue positions. This progress largely addresses the issue from the sequence to the structure supposing the sequence-structure relationship is exclusive, i.e. the amino acid sequence of the protein encodes its unique three-dimensional structure. However, functional binding of proteins is intimately associated with both sequence and structure properties. For instance, very similar protein structures with diverse sequences can dictate different binding specificities with their own partners, and a single sequence can fold in an equilibrium of more than one conformation states which encode different functions[22–25]. In fact, the interaction pattern extracted from the statistical information of MSA is generally common to the whole protein family, but doesn't contain the specific interaction pattern which is unique for a particular functional binding[1,2,26]. For a member of protein family, it is the specific interaction pattern that determines protein's binding specificity to cognate partners and avoids unwanted crosstalks to highly noncognate competitors in the same family[3,27–29]. Therefore, uncovering the full map of common interaction pattern for the whole family and unique interaction pattern for the cognate pairs can better understand the rule of interaction patterns for specific binding.

The interaction pattern of proteins can be finely tuned during evolution to obtain novel function or improve existed function through the process of mutation, adaptation and natural selection. Similarly as Red Queen hypothesis that species must constantly adapt, evolve, and proliferate in order to survive while compete against ever-evolving opposing species[30], proteins at the molecular level also have to optimize the binding specificity with their partners so as to distinguish against binding competitors. A typical binding system between proteins is two-component signaling (TCS) system which is the most prevalent signal transduction system in bacterial for sensing and responding to environment stimuli (Fig. 1)[31,32]. Each bacteria contains tens or hundreds of paralogous TCS. This requires faithful transmission of information between histidine kinases (HKs) and their cognate response regulators (RRs), as well as avoidance of crosstalks[3,28,32]. Previous studies have demonstrated that a small subset of residue positions are critical to the interaction pattern of binding specificity[2,27,29,33]. Substituting residue types of these positions was validated to transfer the binding specificity from cognate to noncognate partners. Meanwhile, increasing evidences have shown that distal positions also affects the specific recognition through intramolecular and intermolecular epistasis[34–37]. How

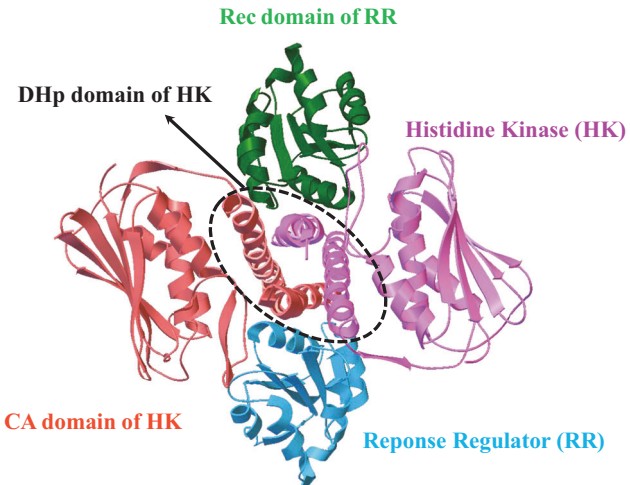

**Fig. 1 Complex structure of two-component signaling system (TCS, PDB entry:3DGE).** TCS contains a cognate protein pair, i.e. histidine kinase (HK) and response regulator (RR); the complex structure of TCS is formed by one HK dimer and two RR monomers; HK dimer is composed of one histidine phosphotransfer (DHp) domain, and two catalytic and ATP-binding (CA) domains; RR is composed of receiver (Rec) domain.

residue positions constitute the interaction patterns for specific binding and how the interaction patterns shaped by evolution are two fundamental questions on binding specificity.

To understand these questions, we carried out a systematic study on the interaction pattern of the typical TCS by combining statistical analysis of evolutionary homologous sequences in nature and physics-based protein evolution simulation at molecular level. The data availability of the homologous sequences and the complex structure of TCS allows us to carry out family-wide evolutionary analysis of natural homologous sequences and structure-oriented evolution simulation. It is found that highly conserved positions cluster at the binding surface for functional recognition. These conserved positions tend to form intramolecular and intermolecular long-range covariation with highly coupling conservations. Positions with highly coupling conservations are sparse but are physically connected through an interaction network. The interaction network provides a family-wide structural basis for long-range modulation of intramolecular folding and intermolecular binding. The unique interaction pattern for specific binding requires further sequence optimization at positions having direct interactions with the cognate partner and those bridging the binding surface and distal regions. Taken together, binding specificity of TCS is determined by both direct intermolecular interactions and long-range epistasis. This work shed light on the rule of how evolution sculpts the interaction patterns for specific binding of TCS.

## Results and discussion

**Position conservations on the binding surface.** In general, globular proteins require folding to form three dimensional structures and binding to perform biological functions. Hydrophobic core of folding is the characteristic to maintain structural stability while functional-binding surface is responsible to directly interact with the partners. Do these two interaction patterns have similar features and what differences are between them? In terms of statistical analysis of MSA, the relationship between hydrophobic preferences and first-order conservation of positions for the Rec domain of RR was investigated. The first-order conservation measures amino acid identity conservation at a given

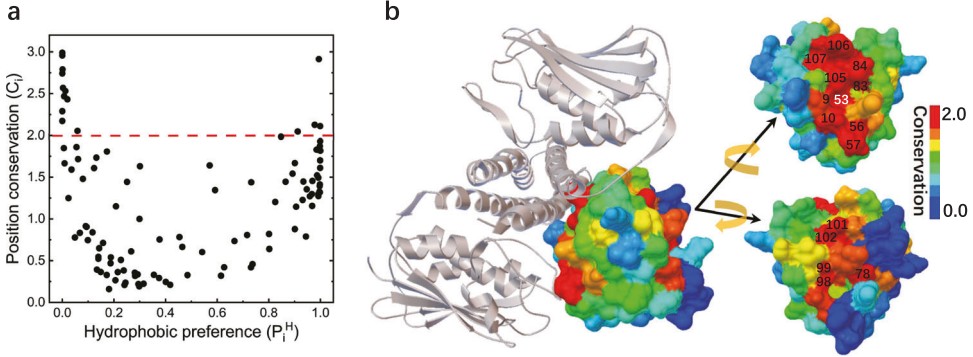

**Fig. 2 Position conservation on the binding surface. a** Position conservation ($C_i$) as a function of hydrophobic preference ($P_i^H$), the red dashed line is to choose the top conserved positions with $C_i > = 2.0$. **b** The top 16 conserved positions (except for position 61 which locates behind position 53) are mapped onto the structure of the Rec domain of RR according to $C_i$ values and HK is shown in gray. Two conserved clusters are separately located on the surface of the Rec domain, one of the clusters locates at the binding surface with upstream cognate HK, and the other one locates at the binding surface of dimerization of phosphorylated Rec domain; the top conserved positions including phosphoacceptor position 53 are labeled.

position, which is expressed through the relative entropy (see the "Methods" section). The relationship manifests two distinct trends (Fig. 2a). The first trend is that the more hydrophobic the positions are, the more conserved they are, while the second trend is that the more hydrophilic the positions are, the more conserved they are.

With the mapping of hydrophobic preference and position conservation respectively on the structures (Supplementary Figs. S1, S2 and Fig. 2b), it can be seen that the positions located in the interior of the Rec domain are both highly hydrophobic and conserved, validating that the hydrophobic core is generally conserved in globular domains[38]. By contrast, on the surface of the Rec domain the majority of the positions are hydrophilic. Those minority hydrophobic positions on the surface mainly participating in the functional binding such as position (14, 54, 56, 84 and 107) at HK-RR interface and position (92, 95, 99 and 102) at dimerization interface of the Rec domain (Supplementary Figs. S1, S2). The distribution of position conservations shows obvious boundary between highly conserved and non-conserved clusters on the molecular surface (Fig. 2b), this is consistent with the separation pattern between functional-binding and non-binding surfaces (Supplementary Fig. S1).

As expected, two phosphoacceptor positions (H260 at the CA domain of HK and D53 at the Rec domain of RR) responsible for auto-phosphorylation, phosphotransfer, and phosphatase activities are most conserved along the sequences (Supplementary Fig. S3). The top 16 conserved positions ($C_i > = 2.0$) of the Rec domain constitute two separate conserved clusters on the structural surface (Fig. 2 and Supplementary Table S1). The first conserved cluster centered at the phosphoacceptor position 53, involves position 9, 10, 56, 57, 61, 83, 84, 105, 106 and 107, suggesting the maintenance and the protection of biological functions from adjacent context shaped by the evolution. Most positions of this conserved cluster participate in the binding surface directly interacting with the DHp and CA domain, such as positions 10, 56, 57, 84, 105, 106 and 107 (Fig. 2b and Supplementary Fig. S1). The second conserved cluster involves position 78, 98, 99, 101 and 102, which locate at the dimerization interface of the Rec domain and are connected by direct contacts or bonds. This conserved cluster may be a common region as a conserved scaffold for the dimerization of the phosphorylated Rec domain in the downstream signaling pathways (Supplementary Fig. S1)[39–41]. Strikingly, all the top conserved positions locate on or near the functional surface rather than hydrophobic core. This could implicate that the interaction pattern in the hydrophobic core required for folding stability is less specific than that for the functional requirement. This separation also

serves to elucidate the relationship of position conservation with specific functional binding in the subsequent discussion. This implicates that functional binding can be more evolutionarily advantageous than structural folding.

**Emergence of coupling conservations for binding**. The organization of protein structure and the adaption of protein function are generally regulated by the intramolecular or intermolecular interaction patterns among residue positions, and optimized though evolution. Local interactions or connected positions tend to constitute the structural context of active sites at the binding surface as discussed above, while the concerted long-range couplings or epistasis between remote positions can modulate conformational dynamics, propagate allostery and alter functions[34–37,42,43]. Numerous studies have demonstrated that epistasis between positions of proteins is extensive and common within protein structures[34–37,42,44]. However, the combinatorial complexity of mapping epistatic effects between positions has severely limited the analysis of the epistatic effect experimentally[45,46]. The epistatic effects as the emergent property of the interaction patterns are imprinted on the native sequences of protein family and optimized by the protein evolution for functional adjustments[47]. Previous studies have demonstrated that statistical coupling extracted from native sequences of protein family is a good indicator of thermodynamic coupling in proteins[34,35]. Coupling conservations provide an implicit way to quantify the epistatic effects emergent from the interaction patterns within the protein structure, and were demonstrated as the major epistatic contributions to the phenotypes of the protein compared to higher-order (≥3) epistatic effects[44,46,48].

We found that the frequency-magnitude relationship of coupling conservations extracted from native sequences follows a power-law distribution within the scale of conservation magnitude (Fig. 3a). In contrast, the frequency-magnitude relationship of the coupling conservation from the random sequences follows a Gaussian distribution and all the values approach zero with no obvious coupling conservations (Supplementary Fig. S4, and Supplementary Dataset S3 and S4). The power-law distribution of non-local coupling conservations suggests that coevolution exists between remote positions within the structure. The coevolution between positions breaks original Gaussian distribution of coupling conservations, and leads to a spacial pattern that a minority of positions with highly coupling conservations prevail over the majority of the positions in long-range communications. This illustrates previous observations that

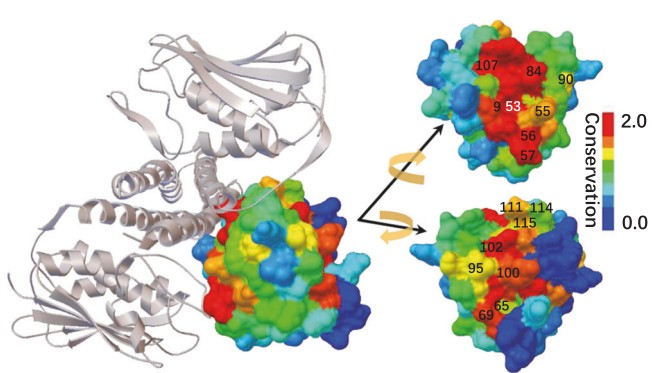

**Fig. 3 Coupling conservation for binding. a** The occurring frequencies as a function of the values of coupling conservation ($C_{ij}$), a power-law-like distribution is observed, the fitting function is $y = a*x^{-b}$, $a = 10^{1.20±0.06}$ and $b = 2.64 ± 0.15$, the R-Square is 0.94. **b** The matrix of coupling conservation with color scaling, highly coupling conservations are colored red ($C_{ij} > = 1.1$). **c** The positions with highly coupling conservations constitute an interaction network physically connected by contacts or bonds. the positions in close proximity to the active site or at the binding surface with HK are grouped in red dotted line, the positions at the dimerization surface of the Rec domain are grouped in blue dotted line, and the positions inside the hydrophobic core are grouped in green dotted line; the contacts or bonds inside the group and between two groups are represented with black and purple lines respectively.

**Fig. 4 19 Positions with highly coupling conservations within Rec domain.** Except for two positions in the hydrophobic core (here not shown), other positions either located in close to the active site or at the binding surface, phosphoacceptor position 53 is also labeled, position 52, 82 (behind 53) and position 94 (behind 95) are not shown, the structural view and color scale are the same as Fig. 2b.

phenotypes of the protein can be represented by a very small number of top contributed epistatic terms[46]. Highly coupling conservations are of remarkable sparsity, which can be an emergent property of evolution at the molecular level (Fig. 3a, b). For example, 51 top-ranked coupling conservations (with $C_{ij} > = 1.10$) out of $172*(172 − 1)/2$ involves only 19 positions in the Rec domain of RR and 3 positions in the DHp domain of HK (Fig. 3b and Supplementary Table S2). All of these 22 positions are relatively conserved having $C_i > = 1.20$ (Supplementary Fig. S3, Supplementary Table S3), suggesting that only conserved positions tend to form highly coupling conservations with each other.

Structurally, those 19 positions with highly coupling conservations in the Rec domain are intensively distributed on the binding surface or near the active site, and inside the hydrophobic core (Figs. 3c, 4 and Supplementary Fig. S1). Also, those 3 positions (259, 262 and 264) in the DHp domain are spatially near the phosphoacceptor/phosphodonor position 260 (Supplementary Fig. S5). As shown in Fig. 5, high couplings occur not only between neighbor positions but also between remote positions. This distribution suggests that coupling conservations play a significant role of long-range communications between the functional binding and the structural folding. In detail, positions with highly coupling conservations in the Rec domain constitute a network spatially connected by the physical interactions (direct contacts or bonds) (Fig. 3c). The sparse but physically connected

network links the positions in close proximity to the active site or at the binding surface with HK, and at the dimerization surface of the Rec domain through the bridging region inside the hydrophobic core (Figs. 3c, 4 and Supplementary Fig. S1). This intramolecular interaction network constitutes long-range communications among the functional-binding surfaces and the hydrophobic core. The coevolving intramolecular interaction network may represent a general interaction pattern that mediates the long-range energetic and dynamic propagation within the proteins[35,37,49].

It is worth noting that the positions (i.e. 55, 69, 90, 94, 100, 102 and 115) participating in high coupling conservations between the Rec domain and the DHp domain are all among those positions which participate in high coupling conservations within the Rec domain (Fig. 3c and Supplementary Table S2). In other words, the positions for highly intermolecular coupling conservations between the Rec domain and the DHp domain are selected from those positions with highly intramolecular coupling conservations of the Rec domain. These highly intermolecular couplings further support that the interaction network of coupling conservation represents a canonical structural basis for energetic and dynamic propagation[34–37,42]. Biological activity is normally modulated not only by direct interactions at the binding surface but also positions across the whole structure. It has been reported that the positions with highly intramolecular coupling conservations are sensitive to modulate protein's binding specificity with its partner[34,50,51]. Taken together, the spatial pattern of sparse coupling conservations are critical to specify protein phenotypes such as folding, binding and long-range allostery.

**Optimization of specific interaction pattern for binding.** Binding specificity between proteins is essential for biological activity in the crowded cellular environment. Understanding how proteins maintain binding specificity to partners and avoid crosstalks to highly similar competitors is a fundamental and challenging issue[52]. Two-component signaling systems rely on the binding specificity to realize precise molecular recognition between the cognate partners and prevent the crosstalk between noncognate pairs[28,28,32]. The statistical information derived from the analysis of MSA is generally common for multiple paralogous TCS, such as position conservations and coupling conservations shown above. It uncovers overlapping properties of diverse TCS but doesn't contain the specific interaction pattern which are unique for a particular cognate pair. The specific interaction pattern depends on the unique sequences and structures of the target cognate protein complex. Recently, we developed a computational structure-oriented evolution simulation method based on the funneled energy landscape theory[50,53]. The simulation of

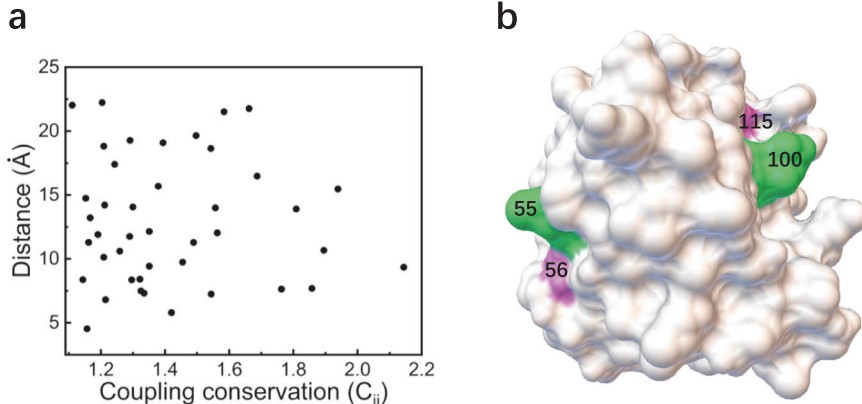

**Fig. 5 Coevolution and coupling between remote positions. a** Scatter plot of coupling distance as a function of the magnitude of coupling conservation ($C_{ij}$). The coupling distance ranges from 4.5 to 22.2 angstrom covers neighbor positions to remote positions. **b** Examples of two remote couplings (position 55–100, and 56–115) were shown, and colored in purple and green respectively on the structure.

protein evolution at the molecular level is to mimic the process of random mutation and selection in nature and search structure-compatible sequences in the sequence space. The resulting evolved sequences carry structure-specific statistical properties and interaction patterns when mapping onto the structure.

To identify structure-specific interaction patterns that confer the binding specificity between cognate partners and prevent crosstalk between noncognate partners, the representative complex of *Thermotoga maritima* class I HK853 and its cognate RR468 is taken as evolutionary target complex[54]. Given the modulation of HK (including the DHp and CA domains) imposed on the evolution of RR (Rec domain) or not, the evolution simulations of the Rec domain were carried out separately under two conditions. i.e. the presence of the DHp and CA domains as the binding partner, and the absence of them respectively (Fig. 1 and Supplementary Methods). Guided by the selection fitness (see the "Methods" section), the evolution dynamics can be visualized as the movements on a projected energy landscape with quantified Shannon entropies of the sequence space and the energies of the target structure (Fig. 6a). The basin of bawl-like evolutionary energy landscape corresponds to the subspace of evolved sequences. The resulting evolved sequences of the Rec domain under two evolution conditions are named as FBSs (folding-binding sequences) and FSs (folding sequences). Similarly as native sequences, the structure-specific interaction patterns formed by the evolved sequences are expected to be in global minimization of frustration.

Previous studies have demonstrated that highly frustrated interactions tend to be clustered on the protein surfaces, and the binding surfaces for functions become less frustrated once specific protein-protein interfaces are formed[55–60]. The quantification of frustration index has been an effective way to analyze the distribution of local frustrations of the whole structure[61,62]. In fact, frustration index can be viewed as the localized quantification of global specificity shaped by the evolution. In order to be consistent with our evolution simulations that use the MJ potential, we have modified the frustratometer algorithm to use the MJ potential instead of the AMW hamiltonian (see Supplementary Methods). We have also validated this modification by comparing these two versions of the frustratometer algorithm for two examples: one is the Rec domain studied here and the other is an example protein in the frustratometer server (details in Supplementary Methods and Dataset S5).

It is observed that the frustration indexes of positions are correlated between FSs and NSs (naturally occurring sequences) with correlation coefficient $R = 0.70$ with $p$-value $< 0.01$ (Fig. 6b).

This high consistence justifies the capability of evolution simulation protocol in generating local interaction patterns of evolved sequences as those of NSs when mapping onto the native structure. Similarly, the frustration indexes were also correlated between FBSs and NSs ($R = 0.52$ with $p$-value $< 0.01$), as well as FBSs and FSs ($R = 0.91$ with $p$-value $< 0.01$)(Fig. 6c, d). The correlations among them could be due to that NSs maintain the common requirements (minimal frustrations) of folding and binding for the family, but lack protein or structure-specific requirements for the binding. Whereas, FSs contain both common and specific requirements for the folding of Rec domain, and FBSs contain common folding/binding requirements and specific binding requirement, as well as most specific folding requirement. Compared to the common requirement for folding, the common requirement for binding could be relatively less in terms of the positions involved. The correlations among them is also simply illustrated by Supplementary Table S4.

The frustration indexes of FBSs were largely changed at 18 positions compared to those of FSs ($|\Delta F_i| > = 0.70$) (Fig. 7d and Supplementary Table S5). The threshold ($=0.70$) was chosen since it separates high frequency peaks representing small frustration changes from low peaks representing large frustration changes (Supplementary Fig. S6). Among these 18 positions, 11 positions become less frustrated and the other 7 positions become more frustrated. The large change of local frustration between FSs and FBSs originates from the presence of binding partner in the evolution simulations. For FBSs, evolved sequences have to adapt to the specific binding interactions in addition to those within the Rec domain by varying the amino acid identities. Energetically, all the positions are globally constrained by the interaction network. Evolution aims to search the interaction patterns which satisfy global minimization of frustration to the large extent by adjusting local frustrations. From the computation equation of local frustration (equation 3 in the "Methods" section), local frustration of the position is determined by the contact energies it has with its surrounding neighbors. Frustration index is quantified by the native energy with respect to the mean value of the decoys, considering the standard deviation from the energy distribution. Taking the position 84 with the largest frustration change as an example (Supplementary Table S5), it is locally frustrated in FSs but minimally frustrated in FBSs. Position 84 has only one contact with position 105 within Rec domain (Supplementary Fig. S7a), thus its local frustration can be largely influenced if additional contacts included. It has additional three contacts (84–260, 84–263 and 84–310) when the specific binding partner HK is present (Supplementary Fig. S7b). These three contacting

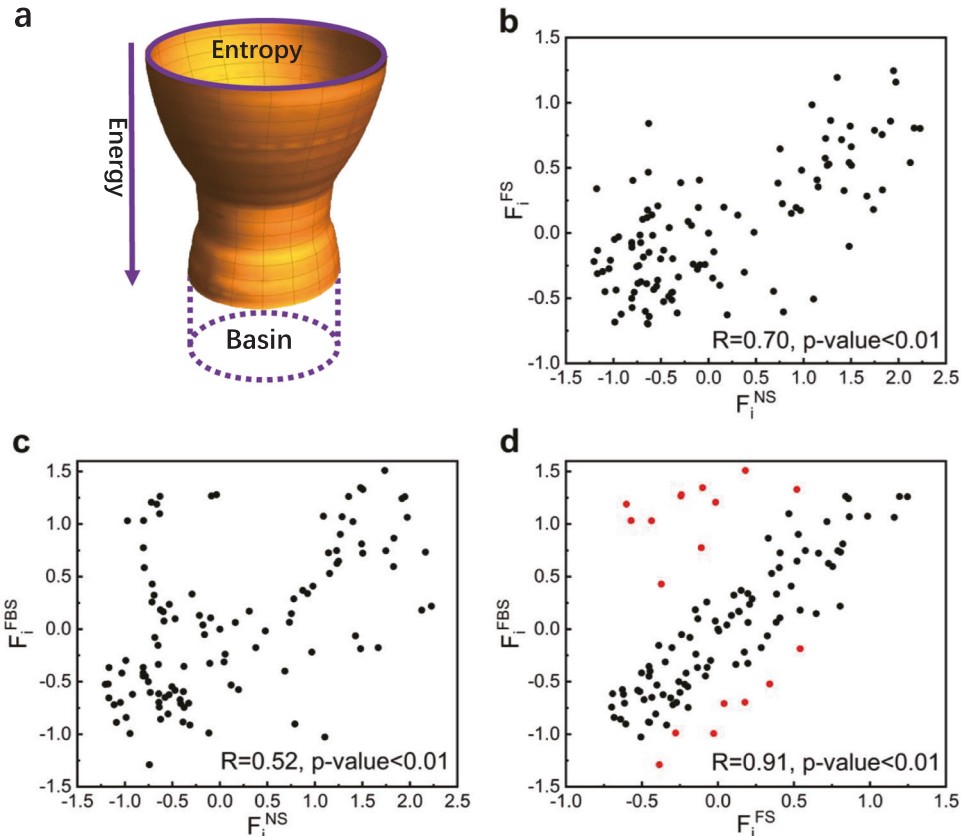

**Fig. 6 Correlation of frustration indexes ($F_i$) among native sequences (NSs), evolved folding sequences (FSs) and folding-binding sequences (FBSs).**
**a** Evolution energy landscape in the sequence space of the Rec domain, the basin means the size of the sequence entropy for the evolved sequences.
**b** Correlation between NSs and FSs, the Pearson correlation coefficient is 0.70 with 2-tailed test of statistical significance $p$-value < 0.01. **c** Correlation between NSs and FBSs, the Pearson correlation coefficient is 0.52 with statistical significance $p$-value < 0.01. **d** Correlation between FSs and FBSs, the Pearson correlation coefficient is 0.91 with statistical significance $p$-value < 0.01, the red points are the positions with $|\Delta F_i| >= 0.70$ between FSs and FBSs.

positions His260, Arg263 and Leu310 all have strong interactions with hydrophobic amino acids according to MJ matrix, which leads to high hydrophobic preference and minimal frustration of position 84 for the evolved sequences in FBSs. Protein evolution balances the folding requirement and functional-binding requirement as energetic conflicts in FSs is largely compensated by the minimal frustrations in FBSs (Fig. 6d)[55,63].

Structurally, these 18 positions either interact with the DHp/CA domain directly or act as the bridging ones between the positions at the binding surface and those across the structure (Fig. 7b). 10 positions locate at the binding surface and have direct interactions between Rec and the DHp/CA domain (Supplementary Fig. S1). This is consistent with the validated experimental observations that position 14, 20 and 21 at the binding surface are crucial to rewire the binding specificity between two different cognate pairs of TCS[27,33]. These 10 evolution-optimized positions are almost complementary to the first highly conserved cluster on the binding surface between the Rec domain and the DHp/CA domain (Figs. 2b and 7b). Together they cover most regions of the whole binding surface. In other words, the functional-binding surface is mainly composed of two classes of positions: the highly conserved positions and structure-specific positions. The highly conserved positions are common for the functional-binding surface at the family-wide level while the structure-specific positions are unique for the members of the family and can be tuned to adapt the cognate partner. By inspecting the relationship between frustration changes and first-order conservation for Rec domain (Fig. 8), it can be seen that only 3 of those positions with highly first-order conservation have large frustration changes ($|\Delta F_i| >= 0.70$) at the

presence of binding partner. This reflects that majority of highly conserved positions at the functional surface satisfy function requirements at family-wide level. In contrast, most of the positions with large frustration changes at the presence of binding partner satisfy structure/protein specific requirements by varying amino acid identities[64].

The other 8 bridging positions link the positions at the binding surface (or near the active site) and the positions across the structure of the Rec domain through the physical contacts/bonds (Supplementary Table S6). For instance, the position 33 contacts with the conserved position 9, 10 and 56 near the active site, and also contacts with the position 15 and 37 which are far and opposite to the binding surface (Supplementary Fig. S8). Researchers have argued that many beneficial mutations are far from the active site and sometimes can not be predicted, or even explained[48,65]. Evolution optimizes the interaction pattern to adapt the binding partner by selecting the amino acid types of the positions not only being located at the binding surface but also the positions bridging the binding surface and distal positions. This provides a possible explanation of how the epistasis between the the remote positions arises. The evolution-optimized structure-specific interaction patterns together with the coupling conservations, provide insight into the explanations and can make predictions on the mutation effects for the positions not at the binding surface.

## Conclusion

Proteins are essential components of living organisms and are involved in a variety of biological processes. Evolution has

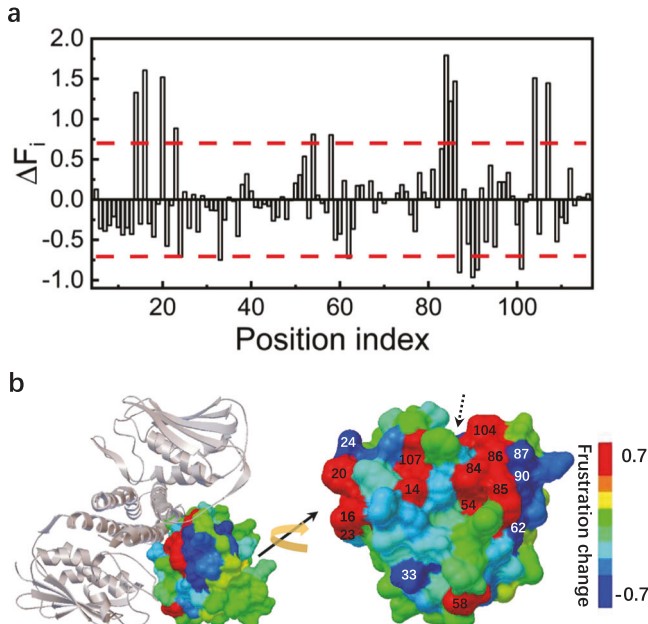

**Fig. 7 Structure-specific interaction patterns optimized by evolution.**
**a** Difference of frustration index between the presence and absence of HK as the evolution template for the Rec domain, the red dashed lines are used to choose the positions which become largely less or more frustrated ($|\Delta F_i| > = 0.7$) when HK is present or not as the template. **b** The positions becoming largely less frustrated are colored in red and labeled in black; while the positions becoming largely more frustrated are colored in blue and labeled in white, including two positions (91 and 101) not shown as the dotted arrow points at.

optimized proteins to form specific interactions between amino acids by selecting sequences and three dimensional structures to satisfy functional requirement and folding stability. Understanding the interaction patterns imprinted on the evolutionary history of protein sequences and structures is a fundamental and challenging issue. In this work, we concentrated on the study of how evolution sculpts interaction patterns that encode the epistasis and binding specificity. We combined statistical analysis of natural homologous sequences to extract family-wide interaction patterns, and structure-oriented evolution simulation to detect structure-specific interaction patterns. By taking TCS as the binding complex, we found three obvious features of the interaction patterns encoded in TCS. First, the amino acid identities at the positions of the functional-binding surface are highly conserved, even more conserved than those at the positions in the hydrophobic core (Fig. 2). This implicates that the interaction pattern in the hydrophobic core required for folding stability may be less specific than that for the functional requirement. Second, the frequency-magnitude relationship of coupling conservations follows power-law-like distribution (Fig. 3a). This supports that under the evolutionary pressure power-law distribution can be an ubiquitous and robust property at different levels in the biological world[66,67]. The positions with highly second-order coupling conservations physically connect to form an interaction network which links functional-binding surfaces and hydrophobic core (Figs. 3b, c, 4 and 5). This suggests that the emergence of highly coupling conservations constitute long-range epistasis between functional binding and structural folding. Third, for the cognate binding structure, additional positions are finely tuned during evolution by minimizing local frustrations at or near the binding surface and sacrificing the stability at other regions (Figs. 6–8). In this way, binding specificity is enhanced and the crosstalk is

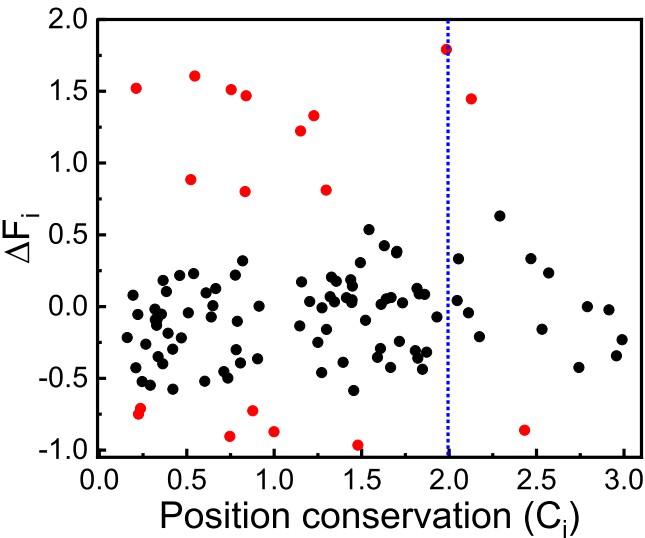

**Fig. 8 Relationship between frustration changes ($\Delta F_i$) and first-order conservation ($C_i$) for positions of Rec domain.** Positions with large frustration change are colored in red and positions with high first-order conservation are separated by blue dotted line ($C_i = 2.0$).

prevented. Taken together, binding specificity of TCS is modulated by both direct interactions and long-range epistasis. The interaction patterns uncovered here provides insight into the rule that governs the epistasis and binding specificity of TCS, and shed lights on the evolutionary design of proteins.

## Methods

**Protein complex model.** Two-component signaling systems (TCS) are the major signal transduction systems in bacterial for sensing and responding to the environment[3,28,31,32]. TCS involve two conserved protein partners which specifically recognize to bind and transfer signal. The TCS partners (histidine kinase (HK) and response regulator (RR)) have mutually evolved to confer specificity which is encoded in the interaction pattern[2,27,29,33]. The cognate signaling and coevolution of HK and RR have made it as a popular protein binding model to study the protein coevolution and binding specificity. HK is composed of two domains: the catalytic and ATPase (CA) domain, and the histidine phosphotransferase (DHp) domain, while RR is generally composed by the receiver (Rec) domain and effector domain (Fig. 1). The effector domain participates in downstream signal transfer, it is not shown in the structure. The DHp domain of HK is responsible for the phosphotransfer to the Rec domain of RR. The native structure of binding complex between *Thermotoga maritima* class I HK853 and its cognate RR468 was taken from the PDB entry 3DGE[54]. Due to C2 symmetry of the complex structure, the binding between HK and RR can be represented by the binding between HK and one Rec domain.

Native homologous sequences of the DHp domain and the Rec domain were taken into account for multiple sequence alignment (MSA). Similar to a lot of other studies[68–71], the standard dataset was taken from the literature[68] which was built by assuming that DHp (Pfam accession ID PF00512, the length is 64) and Rec (Pfam accession ID PF00072, the length is 112) domains adjacent to each other on the genome tend to be cognate pairs with high binding specificity. The sequences with a fraction of gaps greater than 0.2 were removed, which results in 4069 HK/RR pairs of native sequences for MSA (Supplementary Dataset S1). The columns with the fraction of gap amino acids greater than 0.5 were not considered in the computation. In total four positions were deleted and not considered in the computation, they are two

terminal positions of aligned sequences of DHp domain, and one terminal position of aligned sequences of Rec domain. The remaining position is also a gap in the aligned sequence of complex structure. Thus, the aligned sequences are still continuous by mapping them onto the complex structure. Finally, the length of the aligned sequences is 172 which contains 62 positions of DHp domain and 110 positions of Rec domain (Supplementary Dataset S1).

**Quantification of local information.** With MSA, the local information including first-order conservation, hydrophobic preference and local frustration at positions on the sequence can be extracted and mapped onto the native structure. The first-order conservation is computed through Kullback–Leibler divergence (or relative entropy)[72], that is

$$C_i = \sum_{a=1}^{20} f_i^a \ln(f_i^a / q^a) \tag{1}$$

where $f_i^a$ is the observed frequency of residue type a at position i from MSA, there are 20 types for native residues. $q^a$ is the background frequency of residue type a, which is the average occurring frequency in all proteins in the NCBI non-redundant database (Supplementary Table S7)[73]. With the classification of residue hydrophobicity (Supplementary Table S8)[74], the hydrophobic preference of each position on the sequence can be computed as

$$P_i^H = \left(\sum_1^M h_i\right) / M \tag{2}$$

where $h_i$ equals 1 if the residue is hydrophobic, otherwise 0. $M$ is the total number of sequences in MSA. The quantification of local frustration has been an effective way to identify the frustration of a residue position in the whole structure and the residue-level frustration is one type of the local frustrations in terms of the definitions in the reference[55]. It was computed as

$$F_i = \left(<E_i^U> - E_i^N\right) / \sqrt{(1/N)\sum_{k=1}^{N}\left(E_i^U - <E_i^U>\right)^2} \tag{3}$$

where $F_i$ is the frustration index of residue position i, $E_i^N$ is the "native" energy of residue position i. $E_i^U$ is the reference energy of residue position i by randomly selecting the residues occurred in the "native" sequence. Here $N$ (=1000) is the number of randomly selecting times. In the computation, the native conformation of TCS was taken as the target structure. Instead of using the Frustratometer server, the customized code of computing residue-level frustration index was developed and validated (see Supplementary Methods and Supplementary Fig. S9).

**Quantification of coupling information.** Quantification of coupling information between residue positions can be represented by the coupling (second-order) conservation. Statistically, it is computed as

$$C_{ij} = \sqrt{\sum_{a,b} k_i^a k_j^b \left(f_{ij}^{ab} - f_i^a f_j^b\right)^2} \tag{4}$$

where $f_{ij}^{ab}$ is the joint frequency of residue a and b at position i and j respectively, $k_i^a$ (or $k_j^b$) is the coefficient which is the function of the position conservation of residue a at position i, which is $k_i^a = \ln((f_i^a(1-q^a))/(q^a(1-f_i^a)))$. The derivations can be seen in the reference[72].

**Protein evolution principle.** Proteins evolve under the pressure of selection fitness. Our previous studies have suggested that minimal frustration principle of energy landscape can elegantly derive the selection fitness of protein evolution at molecular level

(see Supplementary Methods)[50,53]. According to the derivations, the quantification of selection fitness is represented by the thermodynamic stability and kinetic accessibility of protein folding (or binding) (details in Supplementary Methods). Their expressions are:

$$\Delta G = -K_B T \ln \frac{P_N}{P_D}, \tag{5}$$

and

$$\Lambda = \sqrt{\frac{K_B}{2S} \frac{\delta E}{\Delta E}}. \tag{6}$$

Thermodynamic stability ($\Delta G$) is quantified through the computation of the probabilities of native state and non-native state conformations in the canonical ensemble, i.e. $P_N$ and $P_D$, $K_B$ is Boltzmann constant and T is the temperature. The kinetic accessibility ($\Lambda$) is quantified through the ratio of energy gap $\delta E$ and energy variance $\Delta E$ of the conformation ensemble, $S$ represents conformational entropy (details in Supplementary Methods).

The simulation of protein evolution is to mimic the process of random mutation and selection imposed on the sequence. The fitness function combining both thermodynamic stability and kinetic accessibility determines the selection preference of the sequence in the population. In terms of the expressions, computations of $\Delta G$ and $\Lambda$ require the sampling of conformation ensemble in the structure space (Supplementary Fig. S10).

**Quantification of selection fitness with conformation ensembles.** The fitness function combining both thermodynamic stability and kinetic accessibility determines the selection preference of the sequence in the population. To identify the interaction pattern which are specific for binding, the evolution simulations of Rec domain were carried out separately under two conditions, i.e. the presence of HK as the binding partner and the absence of HK (Supplementary Methods). According to the formations of $\Delta G^f$, $\Lambda^f$, $\Delta G^b$ and $\Lambda^b$ (details in Supplementary Methods), the quantification of them require two sets of conformation ensemble, i.e. the conformation ensemble for the folding of individual Rec domain, and the binding between CA/DHp domain and Rec domain. The native conformations of Rec domain and HK-RR complex were taken from PDB entry 3DGE. Native contact maps of Rec domain as well as its binding with HK is shown in Supplementary Fig. S7. A contact is defined when the distance of any two heavy atoms from two different residues is below a cutoff distance ( = 5.0Å).

The conformation ensemble of folding decoys was constructed by threading the conformations from seven of top 20 abundant families of protein domain (Supplementary Table S9). These seven families were chosen due to their sequence lengths are longer than that of Rec domain. By removing the conformations with gaps and non-standard residue on their sequences, there are 1216 conformations in total as the folding conformation ensemble of Rec domain. For the sequence length consistence with Rec domain, only the first 120 residues were maintained for each conformation. The details of PDB IDs, chain IDs and the starting residue on the sequences are listed in Supplementary Dataset S2. The energy of a decoy conformation for Rec domain is computed as

$$E_1 = \sum_{i,j}^{N=20} \xi(\mu_i, \mu_j)\Delta_{ij} \tag{7}$$

$\xi(\mu_i, \mu_j)$ is the interaction potential of a contact, $\mu_i$ is the type of residue $i$ of 20 natural amino acids. $\Delta_{ij} = 1$ means there is a contact between residue $i$ and $j$, and $\Delta_{ij} = 0$ otherwise. Residue $i$

and $j$ are at least two residue separation. Miyazawa–Jernigan (MJ) matrix (the upper half and diagonal of Table 3 in Ref. [74]) was employed as the interaction potential. MJ potential is a statistical potential built by collecting the frequencies of the residue contact pairs in the native protein structures. It has been widely used in the studies of protein structures, functions and predictions. With the energy distribution of the folding conformation ensemble, $\Delta G^f$ and $\Lambda^f$ can be computed.

The conformation ensemble of binding decoys were generated by docking the Rec domain onto the surface of HK. Molecular docking was carried out with RosettaDock v3.5[75,76]. For the docking between Rec domain and HK (including CA an DHp domains), three steps were performed. First, each docking partner of the complex was prepared in isolation for optimizing their side-chain conformations prior to docking using the prepacking protocol. Second, The prepacked complexes were then relaxed and minimized with high resolution by the refinement protocol. Third, the refined structures were taken as the starting structures for the docking using the local docking perturbation protocol. The smaller protein (i.e. Rec domain) was defined as the docking ligand in the complex and HK was assigned as the receptor which was kept fixed during docking. 2000 ligand orientations were generated by docking. Other docking parameters were set as default. The total energy of a binding complex is computed as

$$E = E_1 + E_2 + E_{12} = \sum_{i,j}^{N=20} \xi(\mu_i, \mu_j)\Delta_{ij} \qquad (8)$$

where $E_1$ and $E_2$ are the intra energies of Rec domain and CA/DHp domains, and $E_{12}$ is the inter energy between Rec domain and CA/DHp domains. Given that the binding is assumed as rigid binding where both Rec domain and CA/DHp domains are fixed in the native conformation. Thus, the total energy can be simplified as $E = E_{12}$ since $E_1$ and $E_2$ are the same for each binding conformation. A contact is formed when the distance of any two heavy atoms from the residue $i$ and $j$ is below a cutoff distance ($= 5.0$Å). With the energy distribution of the binding conformation ensemble, $\Delta G^b$ and $\Lambda^b$ can be readily computed according to their equation 13 and 10 in Supplementary Methods. The energy-conformation relationship shows that the conformations of native binding state are dominant in energetics for the native sequence (from PDB ID 3DGE) no matter Rosetta atomic potentials or residue-level MJ matrix are employed (Supplementary Fig. S10a, c). Also, the energies of the binding conformation ensemble follow a statistical Gaussian-like distribution (Supplementary Fig. S10b, d). This is consistent with the prediction of equation 2 in Supplementary Methods.

**Simulation of structure-oriented protein evolution.** The simulation of protein evolution is to mimic the process of random mutation and selection imposed on the sequence. The evolution in sequence space was simulated with the genetic algorithm. The initial population of sequences were randomly sampled from the sequence space. At the presence of HK, the native structure of binding complex is assumed as the evolved and functional conformation, each sampled sequence must take this native conformation as its unique ground state for folding and binding. At each evolutionary step, one sequence was randomly selected from the population and a random position was mutated for Rec domain. The original sequence was replaced by the mutated sequence if the latter took native structure as its unique ground-state conformation for both Rec domain and the binding complex, otherwise it was replaced by a sequence selected from the population according to the fitness function. The selection fitness of the sequence in the population depends on the probability of rank-based wheel selection, which is $P_{n+1} = P_n(1 - P_n)$, where n

is the rank order of the sequences, it is determined by the sum of the ranks of $\Delta G^f$, $\Lambda^f$, $\Delta G^b$ and $\Lambda^b$ for all the sequences in the population. In this way, the sequences with higher ranks have larger values of fitness, $P_1$ is set to be 0.05. The evolutionary process was repeated until the sequence entropy was convergent. The sequence entropy is the quantitative description of the sequence space during evolution. The sequence entropy of the population along the evolution was computed with Shannon entropy

$$H(S) = \sum_{i=1}^{30} \sum_{j=1}^{20} P_{ij}^S \ln P_{ij}^S \qquad (9)$$

where $j$ is the type of 20 residues and $i$ is the position of the sequence, and $P_{ij}^S$ is the probability of residue type $j$ at position $i$. 10 independent evolutions were performed with different initial population of 500 random sequences. At the absence of HK, the evolution simulation of Rec domain just take folding requirement (i.e. $\Delta G^f$, $\Lambda^f$) as the determinants of the selection fitness. Other parameters are the same.

## Data availability
Supplementary Dataset S1-S5 are available at https://github.com/ZQYanUCAS/EvolutionShapesInteractionPattern, all other data needed to evaluate the conclusions are present in the paper and/or the Supplementary Materials.

## Code availability
The customized codes in the work can be accessed through the link: https://github.com/ZQYanUCAS/EvolutionShapesInteractionPattern.

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

## Acknowledgements

Z.Y. thanks financial supports by Zhejiang Provincial Natural Science Foundation of China (No. LZ24A040002) and start-up grant of Wenzhou Institute, University of Chinese Academy of Sciences (No. WIUCASQD2022019), and computational resources by the High Performance Computing Center of Wenzhou Institute, University of Chinese Academy of Sciences.

## Author contributions

Z.Y. and J.W. designed research; Z.Y. performed research; Z.Y. and J.W. analyzed data; and Z.Y. and J.W. wrote the paper.

## Competing interests

The authors declare no competing interests.
