## [Peer review file · Communications Chemistry]

Evolution shapes interaction patterns for epistasis and specific protein binding in a two-component signaling systemReviewers' comments:

Reviewer #1 (Remarks to the Author):

Yan and Wang proposed a computational investigation to understand how evolution shapes the interaction of a two-component signaling system to maintain specificity. They perform statistical analyses on natural homologous sequences to extract interaction patterns and structure-based evolutionary simulations to isolate structure-specific interaction patterns. They found that binding surfaces are composed of highly conserved positions and structure-specific positions, where the latter can be tuned to adapt to bind the partner.

This is a very interesting work addressing the interplay between protein stability and protein binding affinity in the light of evolution. However, in my opinion, there are some flaws that prevent its publication in the present form. In particular,

- My major concern is that only one case of study is considered but then conclusions are presented as having general validity. The authors should enlarge the dataset by considering other examples of protein-protein interactions and test whether the obtained trends are preserved.
- Coupling conservation frequency is said to follow a power law trend. However, it is hard to say just from looking at Figure 3A. The authors should perform a fit and compare the results against an exponential trend (see for instance the analysis in Figure 3 of <https://doi.org/10.1103/PhysRevE.98.042402>).
- How are insertions and deletions treated in computing the frequencies of Equation 1? Usually, an additional 'gap' amino acid is considered to account for them when computing the frequencies of each amino acid in each column of the MSA.
- The simulation protocol is described without much detail in the main text and all details are in the Supplementary Materials. I suggest adding more details of the followed protocol in the Main text in order to ease the comprehension of the workflow.
- Finally, I recommend careful proofreading of the whole manuscript.

Reviewer #2 (Remarks to the Author):

This article by Zhiqian Yan and Jin Wang investigates the relationship between protein sequence, structure, and function by analyzing how evolutionary forces shape interactions between amino acid positions. Their main results are about conservation of amino acid identities being strong at crucial parts of the protein, i.e. the phosphoacceptor and protein-binding sites; the uncover a power-law-like distribution in the couplings conservation. that connect binding surfaces and the hydrophobic core of the studied proteins; and the emergence of differential frustration signals at specific functional sites when evolving Rec sequences with the authors' already published methodology. Overall, I find the article of interest for the field of proteins biophysics and computational biology. My comments are in the line of improving or adding certain analysis to better support the claims in the article. Finally, as I comment at the end of this revision, the title sounds too general to me, based on the analysis. I would suggest to tone it down to better reflect the findings.

My comments are below:

Minor comments:

Certain vocabulary should be defined better to help the broader reader to understand what they are talking about. An example of this is in page 5: "first order conservation of positions for the Rec domain of RR". It is common to talk about first and second order conservation in the field of DCA and related techniques that are more from the physics side.. A simple description of what first order conservation

means (amino acid identity conservation at a given position, expressed as the sequence entropy) would be enough to avoid readers to be lost in the first part of the results section. A citation to the methods section would also be advisable.

Figures 2, 3, 4: A label for the color bars should be added to allow an easier interpretation of the plots instead of just explaining what the color means in the legend. Same for similar supplementary figures.

The authors should at least briefly justify the use of certain thresholds in their analysis and results. E.g: Why do they use $C_{ij} \geq 1.1$ in figure 3? Why did they use $|\Delta F_{ij}| > 0.7$ in Fig4?

Major comments:

Section: Position conservation at the binding surface.

The authors make a conservation analysis of the amino acid identities within the family MSAs and find that conserved positions are either near the phosphoacceptor site (centered on residue 53) or in the binding surface against DHP and CA domain.

The authors finish the section saying that "Strikingly, all the top conserved positions locate on or near the functional surface rather than hydrophobic core. This implicates that functional binding can be more evolutionarily advantageous than structural folding". I do not agree with the authors regarding this. Conservation is being measured with amino acids being discrete entities where there is no notion of similarity among them. As the hydrophobic interactions at the core of globular proteins do not need to be specific a substitution of one hydrophobic residue by another would not have a big impact on the protein foldability and stability although sequence conservation would be lower if for example half of the proteins have a Leucine in one position and the other half have an Isoleucine. The fact that binding is a more specific type of interaction does not make it "more evolutionarily advantageous". To me that expression sounds like a big statement that is not supported by the data or any other reference in the text. The importance of conservation at higher level phenotypes like frustration show that these two types of interaction can be similarly constrained at the energetic (i.e. frustration) level. See the following preprint to read some examples: <https://doi.org/10.1101/2023.01.25.525527>

Section: Emergence of coupling conservation for binding.

The authors show that "frequency-magnitude relationship of coupling conservations extracted from native sequences follows a power law distribution within the scale of conservation magnitude". They also say that the same does not apply to random sequences. The authors are not explaining how these random sequences are generated and therefore it is not trivial to assess the meaning of that result on random sequences. Are the authors generating random sequences by sampling from natural sequences sort of to maintain some sort of background frequencies for amino acids or any other feature of natural proteins?

The authors say that the data suggest that coevolution exists between remote positions within the structure. I think a scatter plot between the couplings magnitudes and the mean or median distance among residues i and j across the structures of the family would be good to see how true this is beyond a mere suggestion.

The results on figure 3C make sense. I think that adding a color to the circles that reflects the first order conservation scale would help to link both this section with the previous one.

Section: Optimization of Specific Interaction Pattern for Binding.

The authors use a method developed by their group to evolve sequences using natural occurring ones as a starting point. The authors generate evolved sequences for which they thread structures in combination with docking methods.

When the authors calculate frustration do they do it on the monomers separated or in complex with

their partners? Why this is important is discussed in Ferreiro et al 2007 <https://doi.org/10.1073/pnas.0709915104> and Parra et al 2015: <https://doi.org/10.1371/journal.pcbi.1004659>. Just in case, for frustration it is better to perform the analysis with the monomers separated so the frustration signal at the protein-protein binding sites is not compensated. Other than that a delta frustration between frustration calculated both in the monomer and quaternary complexes can be calculated.

For the results in FigS6 the authors say that there is correlation between FSs and NSs. How many NSs there are? They previously mentioned that they started from T. Maritima class I HK853 and its cognate RR468. How many NSs are shown in FigS6. I think the text is not clear and I got lost with that part.

Furthermore the plot in FigS6 shows an important group of residues above the diagonal. An example of this is a point that corresponds to F_i approx 0.4 in FS and -1.1 in NS. That means that the residue is minimally frustrated in the FSs and not in the NSs. I think this is a nice result as it shows that the FSs are more minimally frustrated than the NSs. This is consistent with FSs being simulated without the presence of the partners. I would like to see a similar figure like FigS6 but comparing NSs vs FBSs. They should be more similar (higher correlation) to NSs than FSs. The results are somehow presented in figure 4 but it is not trivial to assess the correlation from it. Maybe also show correlation between FSs and FBSs.

Results in Fig4B are interesting. The rationale would be that if we simulate sequences with and without the evolutionary pressure of binding, the difference in frustration would be attributable to such a requirement.

When the authors say "The frustration indexes of FBSs were significantly altered.." I guess the authors should use another term instead of significantly as no statistical test was used.

A correlation plot showing the relationship between the first order conservation and delta frustration between FSs and FBSs would be useful to support the author's claims in this section. Such plot would be very handy to classify the positions according to family requirements and protein specific requirements.

In general I found that this section is the one that would benefit from further analysis to make the paper stronger. The results are there but not clearly analyzed or shown. Why do positions change in the magnitude and direction observed (red vs blue regions in Fig 4C)? How would the authors biophysically interpret such change? I felt that this is not discussed enough.

Also, the evolution of sequences was made on Rec.. Why not also showing the results if the evolution of sequences is made from the HK sequence? How is frustration at the interactions between HK and Rec? Frustration can be calculated in mutational or configurational modes to make that analysis. Are energies compensated from both sides at the interface or only in one of the two components?

Although the method to localize frustration by Ferreiro et al 2007 is cited, how local frustration calculations were made in practice is not explained. How did the authors calculate frustration for the structures they analyzed? If they used the frustratometer algorithm either the web server or the R package, the authors should cite the corresponding articles where the method is described and the resources made available:

Web server by Parra et al 2016.
<https://doi.org/10.1093/nar/gkw304>

R package by Rausch et al 2021:
doi.org/10.1093/bioinformatics/btab176.

If the authors used custom made scripts to calculate frustration or if they developed their own tool, the code should be provided in a github repository or similar and cited in the methods section so accuracy of the results can be assessed.

Conclusion:

There are some sentences that for me do not add any information like "Evolution has optimized proteins to form specific interactions between amino acids by selecting sequences and three dimensional structures with evolutionary advantages.". Instead of evolutionary advantages maybe they could refer "to satisfy functional, folding and stability requirements".

When they say "First, the positions at the functional-binding surface "is" (this should be "are") highly conserved, even more conserved than the positions at the hydrophobic core" they should refer to "the amino acid identities at the positions...".

References to the figures in the conclusion would be beneficial to help the readers to recapitulate the results in a summarized way.

Summary:

I find the article interesting in general but a more in depth description of the computational experiments would be needed as mentioned in the previous text in order to fully assess the validity of results as well as to be able to reproduce the results by other researchers. The suggested analysis would help to strengthen the results and to support more clearly the authors claims.

Finally, the authors wrote this article based on the analysis of TCSs and therefore I consider that the conclusions cannot be extrapolated in such a general way as the title suggests. I would advise to tone down the title to explicitly say that this has been tested computationally on TCSs.

Importantly. I would suggest the authors upload the raw data and tables as text files to a github repository (or similar) so they can be easily reanalysed by other researchers in the future (including datasets S1 and S2 as well as frustration calculations and the supplementary tables).

Reviewer #1 (Remarks to the Author):

Yan and Wang proposed a computational investigation to understand how evolution shapes the interaction of a two-component signaling system to maintain specificity. They perform statistical analyses on natural homologous sequences to extract interaction patterns and structure-based evolutionary simulations to isolate structure-specific interaction patterns. They found that binding surfaces are composed of highly conserved positions and structure-specific positions, where the latter can be tuned to adapt to bind the partner.

This is a very interesting work addressing the interplay between protein stability and protein binding affinity in the light of evolution. However, in my opinion, there are some flaws that prevent its publication in the present form. In particular,

Response: Thanks for your comments and suggestions. Below are our point-by-point responses to your concerns. The modified texts are marked in red in the revised manuscript.

- My major concern is that only one case of study is considered but then conclusions are presented as having general validity. The authors should enlarge the dataset by considering other examples of protein-protein interactions and test whether the obtained trends are preserved.

Response: Thanks for your concern. In the manuscript, two-component signaling system (TCS) was chosen due to two aspects. First, the complex structure of TCS and the homologous sequences for the cognate pair (i.e. HK and RR) are available in the database. The cognate pair requires the HK and RR are adjacent to each other in the genome so that they are assumed to interact with high specificity. This data availability allows us to carry out family-wide evolutionary analysis of natural homologous sequences and structure-oriented evolution simulation. Second, TCSs are the most prevalent signal transduction systems in bacterial, they are intensively investigated at the experimental side. The experimental observations on the binding specificities had been taken to support and validate our computational and statistical results. The above two aspects match well with our motivation of how evolution shapes the interaction patterns that determine the binding specificity of cognate protein pairs. In the revised manuscript we have toned the title down to “Evolution Shapes Interaction Patterns for Epistasis and Specific Protein Binding: A Computational Test on Two-component Signaling Systems”.

- Coupling conservation frequency is said to follow a power law trend. However, it is hard to say just from looking at Figure 3A. The authors should perform a fit and compare the results against an exponential trend (see for instance the analysis in Figure 3 of <https://doi.org/10.1103/PhysRevE.98.042402>).

Response: Thanks for your insightful suggestions. the figure below shows the fitting curves of the distribution through two fitting functions. One is the power-law distribution, that is $y = ax^{-b}$, b equals 2.64 ± 0.15 and $a = 10^{1.20 \pm 0.06}$; the other is the exponential decay, that is $y = a * \exp(-bx)$, $b = -1.72 \pm 0.11$, $a = 10^{3.28 \pm 0.13}$. The values of fitting parameters and the fitting qualities are listed in the table below. It can be seen that RSS (Residual Sum of Squares) value of power-law fitting is smaller than that of exponential fitting, while the value R-Square of power-law fitting is larger than that of exponential fitting. These fitting results suggest that the distribution of coupling conservation frequency is more like a power-law distribution rather than an exponential one. In the revised manuscript, the fitting curve of power-law distribution has been added in Figure 3a and the fitting function and parameters have been described in the caption.

Model	Power Law		Exponential	
Equation	$y = a * x^{-b}$		$y = a * \exp(-bx)$	
Parameter	$a = 10^{1.20 \pm 0.06}$	$b = 2.64 \pm 0.15$	$a = 10^{3.28 \pm 0.13}$	$b = 1.72 \pm 0.11$
Goodness	RSS=1.46	R-Square=0.942	RSS=1.85	R-Square=0.927

- How are insertions and deletions treated in computing the frequencies of Equation 1? Usually, an additional 'gap' amino acid is considered to account for them when computing the frequencies of each amino acid in each column of the MSA.

Response: Thanks for your concern. A pre-processing was performed for the MSA before the statistical computations. First, as stated in the Methods, the sequences with the fraction of “gap” amino acids greater than 0.2 were removed from the MSA. Second, the columns with the fraction of “gap” amino acids greater than 0.5 were not considered in the computation, i.e. these positions were not analyzed by equation 1-4 in the Methods. In total four positions were deleted and not considered in the computing, they are position 1, 64, 88, 176 (assuming DHP domain of HK is from 1-64, and Rec domain of RR is from 65-176). position 1 and 64 are the terminal positions of aligned sequences

of DHP domain, and position 176 is the terminal position of aligned sequences of Rec domain. In addition, position 88 is also a “gap” amino acid in the aligned sequence of complex structure (PDB ID:3DGE) we employed, Thus, the aligned sequences are still continuous by mapping them onto the complex structure. Finally, the length of the aligned sequences is 172, which contains 62 positions of DHP domain and 110 positions of Rec domain (Supplementary Dataset 1). The above description has been added in the Methods section of revised manuscript.

- The simulation protocol is described without much detail in the main text and all details are in the Supplementary Materials. I suggest adding more details of the followed protocol in the Main text in order to ease the comprehension of the workflow.

Response: Thanks for your suggestion. In the revised manuscript, we have put more details of protein evolution simulation to the Methods section of the main text from Supplemental Materials, including two subsections “Quantification of selection fitness with conformation ensembles” and “Simulation of structure-oriented protein evolution”.

- Finally, I recommend careful proofreading of the whole manuscript.

Response: Thanks for your suggestion. We have carefully proofread the text and figures, and improved the whole manuscript for the revised version. All the modifications are marked in red.

Reviewer #2 (Remarks to the Author):

This article by Zhiqian Yan and Jin Wang investigates the relationship between protein sequence, structure, and function by analyzing how evolutionary forces shape interactions between amino acid positions. Their main results are about conservation of amino acid identities being strong at crucial parts of the protein, i.e. the phosphoacceptor and protein-binding sites; the uncover a power-law-like distribution in the couplings conservation. that connect binding surfaces and the hydrophobic core of the studied proteins; and the emergence of differential frustration signals at specific functional sites when evolving Rec sequences with the authors’ already published methodology. Overall, I find the article of interest for the field of proteins biophysics and computational biology. My comments are in the line of improving or adding certain analysis to better support the claims in the article. Finally, as I comment at the end of this revision, the title sounds too general to me, based on the analysis. I would suggest to tone it down to better reflect the findings.

Response: Thanks for your positive comments and suggestions. We have carefully revised the manuscript according to your comments. As for the title, we have changed it to “Evolution Shapes Interaction Patterns for Epistasis and Specific Protein Binding: A Computational Test on Two-component Signaling System”. Below are our point-by-point responses to your comments. The modified texts are marked in red in the revised

manuscript.

My comments are below:

Minor comments:

Certain vocabulary should be defined better to help the broader reader to understand what they are talking about. An example of this is in page 5: “first order conservation of positions for the Rec domain of RR”. It is common to talk about first and second order conservation in the field of DCA and related techniques that are more from the physics side.. A simple description of what first order conservation means (amino acid identity conservation at a given position, expressed as the sequence entropy) would be enough to avoid readers to be lost in the first part of the results section. A citation to the methods section would also be advisable.

Response: Thanks for your suggestions. In the revised manuscript, we have added the simple description of first-order conservation when it first appears in the Results section. A citation to the Methods section was also stated.

Figures 2, 3, 4: A label for the color bars should be added to allow an easier interpretation of the plots instead of just explaining what the color means in the legend. Same for similar supplementary figures.

Response: Thanks for your helpful suggestion. In the revised manuscript, we have added the labels for the color bars in all related figures. For example, in Figure 3b, we have labeled the color bar by adding “Coupling conservation” alongside the color bar to show the meaning of color scaling. Similar labels were also added in supplemental figures.

The authors should at least briefly justify the use of certain thresholds in their analysis and results. E.g: Why do they use $C_{ij} \geq 1.1$ in figure 3? Why did they use $|\Delta F_i| > 0.7$ in Fig4?

Response: Thanks for your suggestion. the justifications for choosing these thresholds are described below.

For Figure 2a, the threshold of first-order conservation ($C_i = 2.0$) was chosen because that the positions with first-order conservation larger than this threshold all locate at the functional surface rather than the hydrophobic core. This conservation patterns separate functional surface and hydrophobic core, which helps to illustrate the relationship of position conservation with functional binding.

For Figure 3b, the threshold of the coupling conservation ($C_{ij} = 1.1$) was chosen to further illustrate the sparsity of coupling conservations more clearly. This threshold is an example value for illustration, it can take other large values, such as 1.0 or 1.2. Since

the coupling conservation frequencies follow power-law-like distribution. The number of highly coupling conservations is relatively small compared to the total number of pairwise correlation ($(172*(172-1)/2)$, where 172 is the length of aligned sequences. This feature is determined by the power-law distribution (Figure 3a).

For Figure 7a, the threshold of frustration change ($|\Delta F_i|=0.7$) was chosen simply based on the frequency distribution of $|\Delta F_i|$ (see Supplementary Figure 6 in the revised manuscript). It can be seen that besides the high peaks near zero, there are also low peaks. The high peaks and low peaks can be separated by $|\Delta F_i|=0.7$. The high peak means most of the positions have small frustration changes, while those low peaks indicate that certain positions have large frustration changes when the binding is considered in the evolution simulation. Thus, the threshold ($=0.70$) was reasonably chosen for illustrating the frustration in tuning the binding specificity. The description for justifying the use of this threshold was added in the revised manuscript and shown in the Supplementary Figure 6 as below.

Major comments:

Section: Position conservation at the binding surface.

The authors make a conservation analysis of the amino acid identities within the family MSAs and find that conserved positions are either near the phosphoacceptor site (centered on residue 53) or in the binding surface against DHp and CA domain.

The authors finish the section saying that “Strikingly, all the top conserved positions locate on or near the functional surface rather than hydrophobic core. This implicates that functional binding can be more evolutionarily advantageous than structural folding”. I do not agree with the authors regarding this. Conservation is being measured with aminoacids being discrete entities where there is no notion of similarity among them. As the hydrophobic interactions at the core of globular proteins do not need to be

specific a substitution of one hydrophobic residue by another would not have a big impact on the protein foldability and stability although sequence conservation would be lower if for example half of the proteins have a Leucine in one position and the other half have an Isoleucine. The fact that binding is a more specific type of interaction does not make it “more evolutionarily advantageous”. To me that expression sounds like a big statement that is not supported by the data or any other reference in the text. The importance of conservation at higher level phenotypes like frustration show that these two types of interaction can be similarly constrained at the energetic (i.e. frustration) level. See the following preprint to read some examples: <https://doi.org/10.1101/2023.01.25.525527>

Response: Thanks for your insightful comment. Our original expression “functional binding can be more evolutionarily advantageous than structural folding” is obscure. We agree with your statements above.

By inspecting the relationship between frustration changes and first-order conservation for Rec domain (newly added Figure 8), it can be seen that only 3 of these 16 positions with highly first-order conservation have large frustration changes ($|\Delta F_i| \geq 0.7$) at the presence of the binding partner. This reflects that majority of highly conserved positions at the functional surface satisfy function requirement at family-wide level. In contrast, most of the positions with large frustration changes at the presence of binding partner satisfy structure/protein specific requirement by varying amino acid identities. This is consistent with our claim that the highly conserved positions are common for the functional binding surface at the family-wide level while the structure-specific positions are unique for the members of the family and can be tuned through the evolution to adapt the cognate partner.

In the revised manuscript, the above discussion were added to the section “Optimization of Specific Interaction Pattern for Binding”, and the reference you mentioned was cited as Ref. 63.

Section: Emergence of coupling conservation for binding.

The authors show that “frequency-magnitude relationship of coupling conservations extracted from native sequences follows a power law distribution within the scale of conservation magnitude“. They also say that the same does not apply to random sequences. The authors are not explaining how these random sequences are generated and therefore it is not trivial to assess the meaning of that result on random sequences. Are the authors generating random sequences by sampling from natural sequences sort of to maintain some sort of background frequencies for amino acids or any other feature of natural proteins?

Response: Thanks for your question. The random sequences are generated as the following. First, the lengths of the sequences were set as the same as the native sequences, that is 62 positions for DHP domain of HK, and 110 positions for Rec domain of RR. The number of random sequences was set as 5000. Second, for each position of a sequence, the amino acid type was randomly selected from 20 types without any bias, and also the amino acid type of each position is independently assigned. i.e. the amino acids were assigned without any background frequencies or any other feature of natural proteins. The frequency-magnitude relationship of the coupling conservation extracted from these random sequences are near zero and follows Gaussian distribution. This distribution stands in contrast to the power-law distribution

from the native sequences.

The details on generating the random sequences have been added in the caption of Supplementary Figure 4, and the data of 5000 random sequences were uploaded as Supplementary Data 3.

The authors say that the data suggest that coevolution exists between remote positions within the structure. I think a scatter plot between the couplings magnitudes and the mean or median distance among residues i and j across the structures of the family would be good to see how true this is beyond a mere suggestion.

Response: Thanks for your suggestion. As Supplementary Table 2 lists, there are 42 highly coupling conservations within Rec domain (intramolecular). To support that coevolution exists between remote positions in physical distance, the scatter plot between the coupling magnitudes ($C_{ij} \geq 1.1$, the same as Supplementary Table 2) and the mean distance of position i and j have been shown as Figure 5a. The distances ranging from 4.5 angstrom to 22.2 cover neighbor positions to remote positions, suggesting that high couplings occur not only between neighbor positions but also between remote positions. For visualization, two remote couplings (position 55 to 100, and 56 to 115) were shown in the structure (Figure 5b). The supplementary figure have been added in the revised Supplemental Material and shown below.

The results on figure 3C make sense. I think that adding a color to the circles that reflects the first order conservation scale would help to link both this section with the previous one.

Response: Thanks for your helpful suggestion. To be consistent with Figure 2b, we have added a figure as Figure 4 (shown below) which marks these 19 positions with highly coupling conservations within Rec domain. It can be seen that except for the positions in the hydrophobic core, other positions either located in close to the active site or at

the binding surface. The color scale is the same as Figure 2b. Together, the figure clearly support our statement that positions with high first-order conservations (newly added Supplementary Table 3) tend to form highly coupling conservations, and the intramolecular interaction network formed by these positions constitutes long-range communications among functional binding surfaces and hydrophobic core.

Section: Optimization of Specific Interaction Pattern for Binding.

The authors use a method developed by their group to evolve sequences using natural occurring ones as a starting point. The authors generate evolved sequences for which they thread structures in combination with docking methods. When the authors calculate frustration do they do it on the monomers separated or in complex with their partners? Why this is important is discussed in Ferreiro et al 2007 <https://doi.org/10.1073/pnas.0709915104> and Parra et al 2015: <https://doi.org/10.1371/journal.pcbi.1004659>. Just in case, for frustration it is better to perform the analysis with the monomers separated so the frustration signal at the protein-protein binding sites is not compensated. Other than that a delta frustration between frustration calculated both in the monomer and quaternary complexes can be calculated.

Response: Thanks for your question. The evolution simulations of Rec domain were carried out separately under two conditions, i.e. the presence of the DHp and CA domains as the binding partner, and the absence of them. The starting sequences of Rec domain were random sequences (see Methods section) rather than using natural occurring sequences. The resulting evolved sequences of the Rec domain under two evolution conditions are named as FBSs (folding-binding sequences with binding partners) and FSs (folding sequences without binding partners). The calculation of frustration indexes for FSs and FBSs were under the same conditions as their evolution conditions, i.e. the frustration indexes for FSs was performed on the monomer structure (Rec domain) only, while the frustration indexes for FBSs was performed on the

complex structure (Rec domain complex with DHP and CA domains of HK). Thus, Figure 7a reflects the frustration changes (ΔF_i) for Rec domain when it binds to the partner or not. As seen in Figure 7a and Supplementary Figure 6d, the observations are consistent with the discussions in those two literatures you listed. The majority of 18 positions (11/18) with large frustration changes ($|\Delta F_i| \geq 0.70$) becomes less frustrated upon association with the partner, while the remaining (7/18) becomes frustrated. Protein evolution balances the folding requirement and function binding requirement as energetic conflicts in FSs is largely compensated by the minimal frustrations in FBSs. In the revised manuscript, your mentioned reference (Parra et al 2015) has been cited as Ref. 62.

For the results in FigS6 the authors say that there is correlation between FSs and NSs. How many NSs there are? They previously mentioned that they started from T. Maritima class I HK853 and its cognate RR468. How many NSs are shown in FigS6. I think the text is not clear and I got lost with that part.

Response: Thanks for your question. NSs is a set of native homologous sequences which were extracted from Pfam. The number of NSs used in our paper is 4069, which was described in Methods section and listed in Supplementary Dataset 1. The frustration indexes of NSs in the current Figure 6b (original Figure S6) was computed and averaged based on these 4069 sequences.

Our structure-oriented evolution simulation aims to evolve sequences by fixing the target structure. The target structure can be considered as the common structure for NSs, FSs and also FBSs since the flexibility was ignored in our analysis and evolution simulations. The complex structure of T. Maritima class I HK853 and its cognate RR468 was first solved and taken as the target structure in our evolution simulations.

Furthermore the plot in FigS6 shows an important group of residues above the diagonal. An example of this is a point that corresponds to F_i approx 0.4 in FS and -1.1 in NS. That means that the residue is minimally frustrated in the FSs and not in the NSs. I think this is a nice result as it shows that the FSs are more minimally frustrated than the NSs. This is consistent with FSs being simulated without the presence of the partners. I would like to see a similar figure like FigS6 but comparing NSs vs FBSs. They should be more similar (higher correlation) to NSs than FSs. The results are somehow presented in figure 4 but it is not trivial to assess the correlation from it. Maybe also show correlation between FSs and FBSs.

Response: Thanks for your insightful comments. For more clarity, another two correlation plots (NSs vs FBSs, and FSs and FBSs) have been plotted in Figure 6 (shown below). As you stated, there are a few positions which are much more minimally frustrated in FSs than in NSs (Figure 6b).

From the correlation of NSs and FBSs (Figure 6c), we can see that there are more

positions which are frustrated in NSs but minimal frustrated in FBSs. This leads to lower correlation of NSs vs FBSs ($R=0.52$) than NSs vs FSs ($R=0.70$). The lower correlation could be due to that NSs maintain the common requirements (minimal frustrations) of folding and binding for the family, but lack protein or structure specific requirements for the binding. Whereas, FSs contain both common and specific requirements for the folding of Rec domain, and FBSs contain common folding/binding requirements and specific binding requirement, as well as most specific folding requirement. Compared to the common requirement for folding, the common requirement for binding could be relatively less in terms of the positions involved. This also illustrates that the correlation of FSs and FBSs ($=0.91$, Figure 6d) is higher than NSs vs FBSs ($R=0.52$). The correlations among them is also simply described as Supplementary Table4 as below. The above discussion has been added in revised manuscript.

Interaction pattern	Common Folding	Common Binding	Specific Folding	Specific Binding
NSs	Yes	Yes	No	No
FSs	Yes	No	Yes	No
FBSs	Yes	Yes	Yes (mostly)	Yes

Results in Fig4B are interesting. The rationale would be that if we simulate sequences with and without the evolutionary pressure of binding, the difference in frustration would be attributable to such a requirement. When the authors say “The frustration indexes of FBSs were significantly altered..” I guess the authors should use another term instead of significantly as no statistical test was used.

Response: Thanks for your suggestion. We have updated this sentence with “The frustration indexes of FBSs were largely changed...”.

A correlation plot showing the relationship between the first order conservation and delta frustration between FSs and FBSs would be useful to support the author's claims in this section. Such plot would be very handy to classify the positions according to family requirements and protein specific requirements.

Response: Thanks for your insightful suggestion. The scatter plot between the first-order conservation (C_i) and frustration change (ΔF_i) is shown in Figure 8 and also below. The red points are those positions with $|\Delta F_i| \geq 0.7$, the blue dotted line is to separate highly conservation positions with $C_i \geq 2.0$. The figure indicates that those 10 evolution-optimized positions at functional binding surface have 3 overlap positions with highly conserved positions. This supports our claims that those 10 evolution-optimized positions are almost complementary to the first highly conserved cluster on the binding surface between the Rec domain and the DHp/CA domain (Figure 2c). The functional binding surface is mainly composed of two classes of positions: the highly conserved positions for the family requirements and structure-specific positions for protein specific requirements.

In general I found that this section is the one that would benefit from further analysis to make the paper stronger. The results are there but not clearly analyzed or shown.

Why do positions change in the magnitude and direction observed (red vs blue regions in Fig 4C)? How would the authors biophysically interpret such change? I felt that this is not discussed enough.

Response: Thanks for your concern and questions which are really inspiring to further analyze the computation results. Figure 7a shows those 18 positions with large frustration changes between FSs and FBSs. 11 positions becoming largely less frustrated in FBSs compared to FSs are colored in red and labeled in black; while 7 positions becoming largely more frustrated are colored in blue and labeled in white. The values of Delta frustration can be seen in Supplementary Table 5 and also in Figure 6d as below.

The large change of local frustration between FSs and FBSs originates from the presence of binding partner in the evolution simulations. For FBSs, evolved sequences have to adapt to the specific binding interactions in addition to those within the Rec domain by varying the amino acid identities. Energetically, all the positions are globally constrained by the interaction network. Evolution aims to search the interaction patterns which satisfy global minimization of frustration to the large extent by adjusting local frustrations. From the computation equation of local frustration (equation 3 in the main text), local frustration of the position is determined by the contact energies it has with its surrounding neighbors. Frustration index is quantified by the native energy with respect to the mean value of the decoys, considering the standard deviation from the energy distribution. Taking the position 84 with the largest frustration change as an example (Supplementary Table 5), it is locally frustrated in FSs but minimally frustrated in FBSs. Position 84 has only one contact with position 105 within Rec domain (Supplementary Figure 7a), thus its local frustration can be largely influenced if additional contacts included. It has additional three contacts (84-260, 84-263 and 84-310) when the specific binding partner HK is present (Supplementary Figure 7b). These

three contacting positions His260, Arg263 and Leu310 all have strong interactions with hydrophobic amino acids according to MJ matrix, which leads to high hydrophobic preference and minimal frustration of position 84 for the evolved sequences in FBSs.

Also, the evolution of sequences was made on Rec.. Why not also showing the results if the evolution of sequences is made from the HK sequence? How is frustration at the interactions between HK and Rec? Frustration can be calculated in mutational or configurational modes to make that analysis. Are energies compensated from both sides at the interface or only in one of the two components?

Response: Thanks for your concern. As shown in Figure 1, the complex structure of TCS contains HK and RR. While the whole HK is composed of two chains which constitutes one DHP domain and two CA domains. There are two reasons which currently restrict us to study the evolution of HK .

First, only homologous sequences (PF00512) of DHP domain are available in the pfam database, this leads to the statistical analysis being based on the length of DHP domain rather than the whole HK. From the contact map between Rec domain and HK (Supplementary Figure 7b, also shown below), Rec domain not only interacts with DHP domain, but also CA domain. This means that the whole length of HK (including DHP and CA domain) should be considered in the evolution. In other words, the native sequences involve the length of DHP domain while the evolved sequences involve the length of the whole HK. The comparison between them would be incomplete.

Second, due to C2 symmetry of the complex structure, the binding between HK and RR can be represented by the binding between HK and one Rec domain since two Rec domains locate at the opposite surface of HK and have no interactions with each other. Rec domain is an independent evolution and folding unit. This is why we evolve the sequence of one Rec domain at the presence/absence of the whole HK. However, if we evolve the sequence of HK, both two separated Rec domains should be considered simultaneously as binding templates and two binding surfaces between Rec domains and HK should be included in the structure-oriented evolution simulations. In addition, the whole HK contains two chains and three domains, which also involves both folding

and binding. In this case, the evolution of HK at the presence of Rec domains would be driven by folding and binding fitness within HK, as well as binding fitness between Rec domains and HK. This complexity is beyond the reach of our current evolution methods.

Thus, we concentrated on the evolution of Rec domain at the condition with or without the presence of HK.

Although the method to localize frustration by Ferreiro et al 2007 is cited, how local frustration calculations were made in practice is not explained. How did the authors calculate frustration for the structures they analyzed? If they used the frustratometer algorithm either the web server or the R package, the authors should cite the corresponding articles where the method is described and the resources made available:

Web server by Parra et al 2016.

<https://doi.org/10.1093/nar/gkw304>

R package by Rausch et al 2021:

doi.org/10.1093/bioinformatics/btab176.

If the authors used custom made scripts to calculate frustration or if they developed their own tool, the code should be provided in a github repository or similar and cited in the methods section so accuracy of the results can be assessed.

Response: Thanks for your suggestion. As local frustration was firstly quantified in the Ref. 54 (i.e. Ferreiro et al 2007), there are three definitions to compute local frustrations, that is configurational frustration, mutational frustration and residue-level frustration. In our computation, we chose residue-level frustration because it can be compared directly with the position conservation and hydrophobic preference in the analysis. We developed customized codes for computing residue-level frustration with C language due to two aspects. First, in accordance with the residue-level potential (i.e. Miyazawa-Jernigan (MJ) matrix) used in our evolution simulations with residue-level mutation, the MJ matrix rather than AWSEM-based potentials was also employed to compute residue-level frustrations. Second, the residue-level frustration indexes were averaged over a number of native sequences (=4069) or evolved sequences (=5000) mapping onto the target structure (PDB code 3DGE), this requires customized code to carry out processing of multiple sequences rather than once a time on the server.

For referring and assessing our code, we have uploaded it as well as its notes in the github repository (<https://github.com/ZQYan-UCAS/EvolutionShapesInteractionPattern>). We also cited above two references you provided as Ref. 60 and 61 in the texts for introducing the local frustration.

Conclusion:

There are some sentences that for me do not add any information like “Evolution has optimized proteins to form specific interactions between amino acids by selecting sequences and three dimensional structures with evolutionary advantages.“. Instead of evolutionary advantages maybe they could refer “to satisfy functional, folding and stability requirements”.

When they say “First, the positions at the functional-binding surface “is” (this should be “are”) highly conserved, even more conserved than the positions at the hydrophobic core” they should refer to “the amino acid identities at the positions...”.

References to the figures in the conclusion would be beneficial to help the readers to recapitulate the results in a summarized way.

Response: Thanks for your careful reading. We have modified the texts according to your suggestions. We have also cited the figures in the conclusion.

Summary:

I find the article interesting in general but a more in depth description of the computational experiments would be needed as mentioned in the previous text in order to fully assess the validity of results as well as to be able to reproduce the results by other researchers. The suggested analysis would help to strengthen the results and to support more clearly the authors claims.

Response: Thanks for your careful reading and all above insightful comments. Guided by your suggested analysis, the manuscript has been improved to better illustrate the results, more clearly support our claims. In summary, we have newly added Figure 4, Figure 5, Figure 6b and c, Figure 8, and Supplementary Figure 6 and Figure 7; and Supplementary Table 3 and 4, as well as Supplementary Data 3 in the revised Supplemental Material to support our claims in the main text. In addition, to facilitate assessing and reproducing the results by other researchers, we have described the computational details as clearly as possible, moved two Method subsections from the supplemental materials to the main text, and uploaded all the raw data in the gitub repository (<https://github.com/ZQYan-UCAS/EvolutionShapesInteractionPattern>).

Finally, the authors wrote this article based on the analysis of TCSs and therefore I consider that the conclusions cannot be extrapolated in such a general way as the title suggests. I would advise to tone down the title to explicitly say that this has been tested computationally on TCSs.

Response: Thanks for your suggestion, we have toned down the title and change it to” Evolution Shapes Interaction Patterns for Epistasis and Specific Protein Binding: A Computational Test on Two-component Signaling System”. Current title better represents the analysis and findings.

Importantly. I would suggest the authors upload the raw data and tables as text files to a github repository (or similar) so they can be easily reanalysed by other researchers in the future (including datasets S1 and S2 as well as frustration calculations and the supplementary tables).

Response: Thanks for your suggestion. Alongside with the current submission, we have uploaded all the raw data and text files to github repository (the link is: <https://github.com/ZQYan-UCAS/EvolutionShapesInteractionPattern>).

Reviewers' comments:

Reviewer #1 (Remarks to the Author):

The authors addressed my major concerns sufficiently enough. I have just a few minor comments:

- Since the authors opted to restrain their study on a specific class of protein-protein interactions, in my opinion, this choice should be made clearer both in the introduction and conclusion sections. In particular, the last sentence of the introduction should be re-modulated accordingly.
- In Figure 3, the authors should report a measure of how the fit is accurate (e.g. R squared) either in the plot panel or in the figure caption.
- Figure 6, is the P reported in the caption the p-value? In this case, I suggest specifying it in the caption and main text.

Reviewer #2 (Remarks to the Author):

Dear editor,

I am glad to see that the revised manuscript by Zhiqiang Yan and Jin Wang has been substantially improved. I thank the authors for answering my comments with so much detail and effort in their response.

After reading the article again and evaluating their report, I still have the following comments and concerns.

Minor detail:

For all plots that show correlations, the R coefficients, as well as the significance of the correlation (p-values) should be shown in the figure.

Other comments:

The authors respond: "By inspecting the relationship between frustration changes and first-order conservation for Rec domain (newly added Figure 8), it can be seen that only 3 of these 16 positions with highly first-order conservation have large frustration changes ($|\Delta F_i| \geq 0.7$) at the presence of the binding partner. This reflects that majority of highly conserved positions at the functional surface satisfy function requirement at family-wide level. In contrast, most of the positions with large frustration changes at the presence of binding partner satisfy structure/protein specific requirement by varying amino acid identities. This is consistent with our claim that the highly conserved positions are common for the functional binding surface at the family-wide level while the structure-specific positions are unique for the members of the family and can be tuned through the evolution to adapt the cognate partner."

I consider that the way in which the authors have created random sequences is not a fair comparison. By randomizing positions without following background distributions of natural proteins the authors are generating sequences that lie in a completely different sequence space compared to the one of natural, foldable sequences. Because of this, any conclusion drawn from comparing these natural binding proteins to random polymers is not valid, in my opinion.

If the authors want to claim that the power law that they observe is related to the binding function of these proteins, they need to demonstrate that such power law does not exist to non-binding natural

proteins or to random sequences that follow the statistical patterns of natural proteins. Or is this power law a distinctive feature of foldable proteins regardless of them being related to binding? I think the authors should try to answer these questions.

The authors respond: "Thanks for your suggestion. As Supplementary Table 2 lists, there are 42 highly coupling conservations within Rec domain (intramolecular). To support that coevolution exists between remote positions in physical distance, the scatter plot between the coupling magnitudes ($C_{ij} > 1.1$, the same as Supplementary Table 2) and the mean distance of position i and j have been shown as Figure 5a. The distances ranging from 4.5 angstrom to 22.2 cover neighbor positions to remote positions, suggesting that high couplings occur not only between neighbor positions but also between remote positions. For visualization, two remote couplings (position 55 to 100, and 56 to 115) were shown in the structure (Figure 5b). The supplementary figure have been added in the revised Supplemental Material and shown below"

These two examples support the authors' claim. But it is just two pairs. Can the authors add the distances to TableS2 for all pairs of residues so a better evaluation of this claim be assessed?

The authors have made a very convincing explanation (I skip the figure):

"Response: Thanks for your concern. As shown in Figure 1, the complex structure of TCS contains HK and RR. While the whole HK is composed of two chains which constitutes one DHP domain and two CA domains. There are two reasons which currently restrict us to study the evolution of HK. First, only homologous sequences (PF00512) of DHP domain are available in the pfam database, this leads to the statistical analysis being based on the length of DHP domain rather than the whole HK. From the contact map between Rec domain and HK (Supplementary Figure 7b, also shown below), Rec domain not only interacts with DHP domain, but also CA domain. This means that the whole length of HK (including DHP and CA domain) should be considered in the evolution. In other words, the native sequences involve the length of DHP domain while the evolved sequences involve the length of the whole HK. The comparison between them would be incomplete. Second, due to C2 symmetry of the complex structure, the binding between HK and RR can be represented by the binding between HK and one Rec domain since two Rec domains locate at the opposite surface of HK and have no interactions with each other. Rec domain is an independent evolution and folding unit. This is why we evolve the sequence of one Rec domain at the presence/absence of the whole HK. However, if we evolve the sequence of HK, both two separated Rec domains should be considered simultaneously as binding templates and two binding surfaces between Rec domains and HK should be included in the structure-oriented evolution simulations. In addition, the whole HK contains two chains and three domains, which also involves both folding 15 / 17 and binding. In this case, the evolution of HK at the presence of Rec domains would be driven by folding and binding fitness within HK, as well as binding fitness between Rec domains and HK. This complexity is beyond the reach of our current evolution methods. Thus, we concentrated on the evolution of Rec domain at the condition with or without the presence of HK."

I consider that such explanation should be included as a supplementary note and cited briefly in the main text so readers can better understand in case they have the same question as I had.

The authors respond: "Response: Thanks for your suggestion. As local frustration was firstly quantified in the Ref. 54 (i.e. Ferreiro et al 2007), there are three definitions to compute local frustrations, that is configurational frustration, mutational frustration and residue-level frustration. In our computation, we chose residue-level frustration because it can be compared directly with the position conservation and hydrophobic preference in the analysis. We developed customized codes for computing residue-level frustration with C language due to two aspects. First, in accordance with the residue-level potential (i.e. Miyazawa-Jernigan (MJ) matrix) used in our evolution simulations with residue-level mutation, the MJ matrix rather than AWSEM-based potentials was also employed to compute residue-level frustrations. Second, the residue-level frustration indexes were averaged over a number of native sequences (=4069) or evolved sequences (=5000) mapping onto the target structure (PDB code 3DGE), this requires customized code to carry out processing of multiple sequences rather than

once a time on the server. For referring and assessing our code, we have uploaded it as well as its notes in the github repository (<https://github.com/ZQYanUCAS/EvolutionShapesInteractionPattern>). We also cited above two references you provided as Ref. 60 and 61 in the texts for introducing the local frustration.”.

The use of the MJ energy function in the authors' modified version of the frustratometer algorithm is indeed very interesting. Thanks for clarifying. I think it would be useful if the authors provide for some cases a comparative between the frustration results from the Ferreiro group (AWSEM hamiltonian) and compare it with the MJ one that they developed. Are the results very different? If yes, where? A scatter plot for the residue level frustration index, using both versions, for the reference structures could be a way to show how they compare to each other.

The reason for this is that the frustration algorithm has been available for a long time, and has been widely cited, and to use a modified algorithm sounds strange unless there is a clear reason, which the authors seem to have. The authors should clearly mention in their main text that frustration was calculated with a modified version of the original algorithm and state the reasons for this (which were explained in the above answer) very clearly. The comparison that I am asking is fundamental to provide credibility to the study and not to present it as an ad-hoc decision.

Summary:

I find that the manuscript has been really improved. However, in my opinion, certain parts need to be clarified a bit more in order to fully present the results in a reproducible manner.

Reviewer #1 (Remarks to the Author):

The authors addressed my major concerns sufficiently enough. I have just a few minor comments:

Response: Thank you for all your previous concerns on our work. Below are our point-by-point responses to your new comments. The modified texts are marked in red in the revised manuscript.

- Since the authors opted to restrain their study on a specific class of protein-protein interactions, in my opinion, this choice should be made clearer both in the introduction and conclusion sections. In particular, the last sentence of the introduction should be re-modulated accordingly.

Response: Thanks for your suggestion. In the revised manuscript, we have added the rationale of choice in the introduction and restrained the conclusions on TCS only.

- In Figure 3, the authors should report a measure of how the fit is accurate (e.g. R squared) either in the plot panel or in the figure caption.

Response: Thanks for your suggestion. We have put the value of R-Square for the power-law fitting in the caption of Figure 3.

- Figure 6, is the P reported in the caption the p-value? In this case, I suggest specifying it in the caption and main text.

Response: Yes, it is the p-value. The p-value is computed with 2-tailed test of significance. Since the p-values are very small (smaller than 1.0E-8) for the correlation coefficients of Figure 6, thus p-value<0.01 is stated in the caption. We also stated them in the figures and in the main text.

Reviewer #2 (Remarks to the Author):

Dear editor,

I am glad to see that the revised manuscript by Zhiqiang Yan and Jin Wang has been substantially improved. I thank the authors for answering my comments with so much detail and effort in their response.

After reading the article again and evaluating their report, I still have the following comments and concerns.

Response: Thank you for all your previous comments and concerns. Below are our point-by-point responses to your new comments. The modified texts are marked in red

in the revised manuscript.

Minor detail:

For all plots that show correlations, the R coefficients, as well as the significance of the correlation (p-values) should be shown in the figure.

Response: Thanks for your suggestion. Since the p-values are very small (smaller than $1.0E-8$) for the correlation coefficients of Figure 6 and newly added Figure S9, thus $p\text{-value} < 0.01$ are stated.

Other comments:

The authors respond: “By inspecting the relationship between frustration changes and first-order conservation for Rec domain (newly added Figure 8), it can be seen that only 3 of these 16 positions with highly first-order conservation have large frustration changes ($|\Delta F_i| \geq 0.7$) at the presence of the binding partner. This reflects that majority of highly conserved positions at the functional surface satisfy function requirement at family-wide level. In contrast, most of the positions with large frustration changes at the presence of binding partner satisfy structure/protein specific requirement by varying amino acid identities. This is consistent with our claim that the highly conserved positions are common for the functional binding surface at the family-wide level while the structure-specific positions are unique for the members of the family and can be tuned through the evolution to adapt the cognate partner.”

I consider that the way in which the authors have created random sequences is not a fair comparison. By randomizing positions without following background distributions of natural proteins the authors are generating sequences that lie in a completely different sequence space compared to the one of natural, foldable sequences. Because of this, any conclusion drawn from comparing these natural binding proteins to random polymers is not valid, in my opinion.

If the authors want to claim that the power law that they observe is related to the binding function of these proteins, they need to demonstrate that such power law does not exist to non-binding natural proteins or to random sequences that follow the statistical patterns of natural proteins. Or is this power law a distinctive feature of foldable proteins regardless of them being related to binding? I think the authors should try to answer these questions.

Response: Thanks for your concern. The power law distribution of coupling conservations (Figure 3a) is extracted from natural homologous sequences including the whole length of both the Rec domain and DHP domain, i.e. the intramolecular and intermolecular couplings are both considered. The case is the same if only natural homologous sequences of Rec domain is considered for plotting the figure (see the

figure below). As you speculated, the power law distribution of coupling conservations could be a distinctive feature of foldable proteins (single chain domain or multichain complex) against random sequences.

To further validate the comparison between random sequences and natural sequences, we have generated alternative random sequences with the background distributions of amino acids in natural proteins (Table S7), let's call them RanSeqB (newly added Supplementary Dataset 4). The frequency-magnitude relation of the coupling conservation (see the figure below) still follows Gaussian-like distribution as Figure S4 which is from totally random sequences (let's call them as RanSeqA). The differences between these two kinds of random sequences (RanSeqB and RanSeqA) is that the values of coupling conservation from RanSeqB are more approaching zero, this is because the coupling conservation was estimated by eliminating the effect from background distributions of amino acids (see equation 1 and 4 in the main text). Together, these two figures further support our claims in the main text. We have merged Figure S4 with the below figure as the new Figure S4 in the revised manuscript.

The authors respond: "Thanks for your suggestion. As Supplementary Table 2 lists, there are 42 highly coupling conservations within Rec domain (intramolecular). To support that coevolution exists between remote positions in physical distance, the scatter plot between the coupling magnitudes ($C_{ij} \geq 1.1$, the same as Supplementary Table 2) and the mean distance of position i and j have been shown as Figure 5a. The distances ranging from 4.5 angstrom to 22.2 cover neighbor positions to remote positions, suggesting that high couplings occur not only between neighbor positions but also between remote positions. For visualization, two remote couplings (position 55 to 100, and 56 to 115) were shown in the structure (Figure 5b). The supplementary figure have been added in the revised Supplemental Material and shown below"

These two examples support the authors' claim. But it is just two pairs. Can the authors add the distances to TableS2 for all pairs of residues so a better evaluation of this claim be assessed?

Response: Thanks for your suggestion. In the revised manuscript, we have added the distances of those top coupling pairs within Rec domain in a new column of Table S2.

The authors have made a very convincing explanation (I skip the figure):

"Response: Thanks for your concern. As shown in Figure 1, the complex structure of TCS contains HK and RR. While the whole HK is composed of two chains which constitutes one DHP domain and two CA domains. There are two reasons which

currently restrict us to study the evolution of HK. First, only homologous sequences (PF00512) of DHP domain are available in the pfam database, this leads to the statistical analysis being based on the length of DHP domain rather than the whole HK. From the contact map between Rec domain and HK (Supplementary Figure 7b, also shown below), Rec domain not only interacts with DHP domain, but also CA domain. This means that the whole length of HK (including DHP and CA domain) should be considered in the evolution. In other words, the native sequences involve the length of DHP domain while the evolved sequences involve the length of the whole HK. The comparison between them would be incomplete. Second, due to C2 symmetry of the complex structure, the binding between HK and RR can be represented by the binding between HK and one Rec domain since two Rec domains locate at the opposite surface of HK and have no interactions with each other. Rec domain is an independent evolution and folding unit. This is why we evolve the sequence of one Rec domain at the presence/absence of the whole HK. However, if we evolve the sequence of HK, both two separated Rec domains should be considered simultaneously as binding templates and two binding surfaces between Rec domains and HK should be included in the structure-oriented evolution simulations. In addition, the whole HK contains two chains and three domains, which also involves both folding and binding. In this case, the evolution of HK at the presence of Rec domains would be driven by folding and binding fitness within HK, as well as binding fitness between Rec domains and HK. This complexity is beyond the reach of our current evolution methods. Thus, we concentrated on the evolution of Rec domain at the condition with or without the presence of HK.”

I consider that such explanation should be included as a supplementary note and cited briefly in the main text so readers can better understand in case they have the same question as I had.

Response: Thanks for your suggestion. We have added this explanation fully in the Supplementary methods with the subsection “Setup for protein binding and evolution of TCS”, also the subsection has been cited briefly in the corresponding places of the main text.

The authors respond: “Response: Thanks for your suggestion. As local frustration was firstly quantified in the Ref. 54 (i.e. Ferreiro et al 2007), there are three definitions to compute local frustrations, that is configurational frustration, mutational frustration and residue-level frustration. In our computation, we chose residue-level frustration because it can be compared directly with the position conservation and hydrophobic preference in the analysis. We developed customized codes for computing residue-level frustration with C language due to two aspects. First, in accordance with the residue-level potential (i.e. Miyazawa-Jernigan (MJ) matrix) used in our evolution simulations with residue-level mutation, the MJ matrix rather than AWSEM-based potentials was also employed to compute residue-level frustrations. Second, the residue-level frustration indexes were averaged over a number of native sequences (=4069) or evolved sequences (=5000) mapping onto the target structure (PDB code 3DGE), this

requires customized code to carry out processing of multiple sequences rather than once a time on the server. For referring and assessing our code, we have uploaded it as well as its notes in the github repository (<https://github.com/ZQYanUCAS/EvolutionShapesInteractionPattern>). We also cited above two references you provided as Ref. 60 and 61 in the texts for introducing the local frustration.”.

The use of the MJ energy function in the authors' modified version of the frustratometer algorithm is indeed very interesting. Thanks for clarifying. I think it would be useful if the authors provide for some cases a comparative between the frustration results from the Ferreiro group (AWSEM hamiltonian) and compare it with the MJ one that they developed. Are the results very different? If yes, where? A scatter plot for the residue level frustration index, using both versions, for the reference structures could be a way to show how they compare to each other.

The reason for this is that the frustration algorithm has been available for a long time, and has been widely cited, and to use a modified algorithm sounds strange unless there is a clear reason, which the authors seem to have. The authors should clearly mention in their main text that frustration was calculated with a modified version of the original algorithm and state the reasons for this (which were explained in the above answer) very clearly. The comparison that I am asking is fundamental to provide credibility to the study and not to present it as an ad-hoc decision.

Response: Thanks for your insightful suggestion. In the revised manuscript, we have compared the frustration indexes at residue level for two cases. One is the Rec domain of TCS (PDB ID: 3DGE, chain C) which we employed as the protein model, the other is an example protein (PDBID: 4ZKQ, chain A) which is employed as one of the protein models to show the results of frustration indexes in the Frustratometer server (<http://frustratometer.qb.fcen.uba.ar>). The frustration index results of these two cases computed with AMW hamiltonian on the server can be obtained from the links (<http://frustratometer.qb.fcen.uba.ar/results/201637193147661102>, <http://bonarda.qb.fcen.uba.ar/results/202312734041952360>) and the residue-level frustration indexes are shown in the newly added Supplementary dataset 5. The residue-level frustration index results computed by our customized code with MJ hamiltonian were also shown in the newly added Supplementary Dataset 5.

It can be seen that the correlation coefficients between two versions for these two cases are 0.66 and 0.65 respectively (newly added Figure S9, and also shown below). These correlation coefficients suggest that majority of the frustration indexes from two versions are consistent. By mapping the differences of frustration indexes ($\Delta F = |F_{\text{custom}} - F_{\text{server}}|$) onto the structure (Figure S9e and f), we found that the positions with large differences mainly located at the structural surfaces, especially the loop regions on the surfaces. In detail, if we set a threshold ($\Delta F > 1.6$) for choosing the positions with large differences between two versions, for the Rec domain, these

positions are intensively located at the loop regions (position 43 to 45 and position 70 to 76). These positions have no overlaps with those which were identified as important positions for functional binding of the Rec domain (Figure 2, 3, 4 and 7). In other words, the observed results from our customized and modified version of calculating residue-level frustration index are robust and wouldn't be changed if the original version with AMW hamiltonian were employed. The difference between two versions could be due to that AMW potentials contains not only pairwise contact, but also a single-residue burial term which accounts for the solvent exposure of the residue. The loop regions on the structural surface generally have few contacts and exposed to solvent. In this sense, AMW potentials could more accurate to represent the surrounding context of the residue positions at loop regions.

We have added above clarity in the Supplementary Methods as a subsection “Computation of Residue-level frustration index with Customized code”, and cited briefly in the main text.

Summary:

I find that the manuscript has been really improved. However, in my opinion, certain parts need to be clarified a bit more in order to fully present the results in a reproducible manner.

Response: Thanks for all your insightful suggestions and comments. We hope the revised version of our manuscript has been clarified clearly.

REVIEWERS' COMMENTS:

Reviewer #2 (Remarks to the Author):

Dear editor,

I find that the authors have satisfactorily answered most of my concerns and I am glad to see that the manuscript reads much better and precise and, in my opinion, quite improved.

I still have one concern. The authors have made a good explanation of the differences between their customized version of the frustratometer algorithm that uses the MJ energy function and the original one that uses the AWSEM energy function. However, there is no mention of the use of a modified version of the frustratometer algorithm, in which the authors use the MJ potential instead of the original one in order to be consistent with their own simulation tools. This needs to be clarified in the main text because as it is now, only readers that go to the supplementary methods will realize this. As the authors explained, there are differences between the two versions, especially in the loop regions. In order to avoid confusions, the main text needs to include this clarification and perhaps the supplementary section that explains this as a supplementary method, should be included in the main methods as this is central in their study. Sorry to insist on this but I think that it is important. Still, I think that using a different potential in the algorithm is a very interesting thing to do.

A good way to clarify this would be to do it in page 14 where they first make a small introduction about frustration. I would do it as follows:

"The quantification of frustration index has been an effective way to analyze the distribution of local frustrations of the whole structure (61, 62). In fact, frustration index can be viewed as the localized quantification of global specificity shaped by the evolution. EXPLANATION HERE. It is observed that the frustration indexes of positions are correlated between FSs and NSs (naturally occurring sequences) with correlation coefficient $R = 0.70$ with $p\text{-value} < 0.01$ (Figure 6b).

Explanation could be something in the line of: "In order to be consistent with our evolution simulations that use the MJ potential we have modified the frustratometer algorithm to use the MJ energy function instead of the AWSEM one (see methods).

I have no further comments. I wish the authors a good end of year.

Best regards,

R. Gonzalo Parra

Reviewer #2 (Remarks to the Author):

Dear editor,

I find that the authors have satisfactorily answered most of my concerns and I am glad to see that the manuscript reads much better and precise and, in my opinion, quite improved.

I still have one concern. The authors have made a good explanation of the differences between their customized version of the frustratometer algorithm that uses the MJ energy function and the original one that uses the AWSEM energy function. However, there is no mention of the use of a modified version of the frustratometer algorithm, in which the authors use the MJ potential instead of the original one in order to be consistent with their own simulation tools. This needs to be clarified in the main text because as it is now, only readers that go to the supplementary methods will realize this. As the authors explained, there are differences between the two versions, especially in the loop regions. In order to avoid confusions, the main text needs to include this clarification and perhaps the supplementary section that explains this as a supplementary method, should be included in the main methods as this is central in their study. Sorry to insist on this but I think that it is important. Still, I think that using a different potential in the algorithm is a very interesting thing to do.

A good way to clarify this would be to do it in page 14 where they first make a small introduction about frustration. I would do it as follows:

“The quantification of frustration index has been an effective way to analyze the distribution of local frustrations of the whole structure (61, 62). In fact, frustration index can be viewed as the localized quantification of global specificity shaped by the evolution. EXPLANATION HERE. It is observed that the frustration indexes of positions are correlated between FSs and NSs (naturally occurring sequences) with correlation coefficient $R = 0.70$ with $p\text{-value} < 0.01$ (Figure 6b).

Explanation could be something in the line of: “In order to be consistent with our evolution simulations that use the MJ potential we have modified the frustratometer algorithm to use the MJ energy function instead of the AWSEM one (see methods).

I have no further comments. I wish the authors a good end of year.

Best regards,

R. Gonzalo Parra

Response: Gonzalo, thank you for your patience and consideration during this wonderful discussion journey. As for your last concern, we have followed your kindly suggestion to improve the readability of our manuscript. We have added the paragraph below in the main text, and kept the details in Supplementary Methods”

“In order to be consistent with our evolution simulations that use the MJ potential, we have modified the frustratometer algorithm to use the MJ potential instead of the AMW hamiltonian (see Supplementary Methods). We have also validated this modification by comparing these two versions of the frustratometer algorithm for two examples: one is the Rec domain studied here and the other is an example protein in the frustratometer server (details in Supplementary Methods).”